# TAMI: Taming Heterogeneity in Temporal Interactions for Temporal Graph Link Prediction

**Zhongyi Yu**[1], **Jianqiu Wu**[1], **Zhenghao Wu**[2], **Shuhan Zhong**[3],
**Weifeng Su**[1,5], **Chul-Ho Lee**[4], **Weipeng Zhuo**[1,5*]

[1]Beijing Normal-Hong Kong Baptist University, [2]University College Dublin
[3]The Hong Kong University of Science and Technology, [4]Texas State University
[5]Guangdong Provincial / Zhuhai Key Laboratory of IRADS, China
{zhongyiyu,jianqiuwu,wfsu,weipengzhuo}@bnbu.edu.cn,
zhenghao.wu@ucdconnect.ie, szhongaj@cse.ust.hk, chulho.lee@txstate.edu

## Abstract

Temporal graph link prediction aims to predict future interactions between nodes in a graph based on their historical interactions, which are encoded in node embeddings. We observe that heterogeneity naturally appears in temporal interactions, e.g., a few node pairs can make most interaction events, and interaction events happen at varying intervals. This leads to the problems of ineffective temporal information encoding and forgetting of past interactions for a pair of nodes that interact intermittently for their link prediction. Existing methods, however, do not consider such heterogeneity in their learning process, and thus their learned temporal node embeddings are less effective, especially when predicting the links for infrequently interacting node pairs. To cope with the heterogeneity, we propose a novel framework called TAMI, which contains two effective components, namely log time encoding function (LTE) and link history aggregation (LHA). LTE better encodes the temporal information through transforming interaction intervals into more balanced ones, and LHA prevents the historical interactions for each target node pair from being forgotten. State-of-the-art temporal graph neural networks can be seamlessly and readily integrated into TAMI to improve their effectiveness. Experiment results on 13 classic datasets and three newest temporal graph benchmark (TGB) datasets show that TAMI consistently improves the link prediction performance of the underlying models in both transductive and inductive settings. Our code is available at `https://github.com/Alleinx/TAMI_temporal_graph`.

## 1 Introduction

Temporal link prediction is a fundamental task that forecasts future interactions between two nodes on continuous-time temporal graphs (CTTGs). This is important in a variety of real-world scenarios with dynamic graph topologies changing over time, such as social networks [24, 1], user-item interaction systems [13, 41, 38, 40], traffic networks [35, 32, 37], and physical systems [23, 20]. The temporal link prediction task involves two major steps. The first step is to compute the temporal embedding of each node by aggregating information from its own historical interactions with neighboring nodes. For each target node pair for link prediction, the second step is to take their embeddings into a link predictor (typically an MLP) to estimate the probability of having a link between them.

Heterogeneity arises in temporal interactions. For example, the number of interactions for each node pair varies substantially. The time between two consecutive interaction events can also be

---

[*]Corresponding author.

quite different. To see this, in Figure 1, we plot the distribution of interaction intervals between any pair of nodes on the UCI dataset, showing that it follows a power-law distribution. In other words, most interactions are frequent ones, while the *infrequent* ones (very large interaction intervals) still exist with a non-negligible probability. We also compute and report the Fisher's skewness in Figure 1, indicating that the distribution is positively skewed or right-skewed. Here the skewness $\Gamma$ of a random variable $X$ is defined as $\Gamma = \mathbb{E}\left[(X - \mu)^3/\sigma^3\right]$, where $\mu$ and $\sigma$ denote the mean and standard deviation of $X$, respectively.

The heterogeneity in temporal interactions poses two primary challenges to existing methods for temporal link prediction [34, 7, 39]. First, they use sinusoidal functions as their time encoding functions to encode the temporal difference between the target time $\tau$ and each interaction time $t_i$. We observe that the frequency parameters of the time encoding functions are *harder to learn* when the distribution of the temporal difference is highly skewed, which is the common case as seen from the distribution of interaction intervals. Second, they learn the embedding of each node based only on its recent interaction events. While the embedding encodes timely information, it can be predominantly influenced by a few neighbors having frequent interactions. As a result, the temporal link prediction can be inaccurate for a target node pair having *infrequent* interactions over time. For example, it is natural to predict that a couple would eat hamburgers if it is one of their recent favorites (frequent interactions). However, it makes more sense to predict that turkey will be on the table (infrequent interactions) if Thanksgiving is coming.

To address the above challenges, we propose TAMI, a novel framework that **tam**es the heterogeneity in temporal **i**nteractions to improve the performance of temporal link prediction. TAMI mainly contains two novel components, namely log time encoding function (LTE) and link history aggregation (LHA). We propose to use a logarithmic transformation in LTE to rescale the temporal difference before it is taken into the time encoding function, so that the frequency parameters of the time encoding function can be easier (or quicker) to learn. In addition, we develop LHA to preserve the information of the most recent $k$ interactions for each target node pair for link prediction, regardless of their interaction frequency. In other words, the historical interactions of each node pair are explicitly used for their link prediction, no matter when they happened.

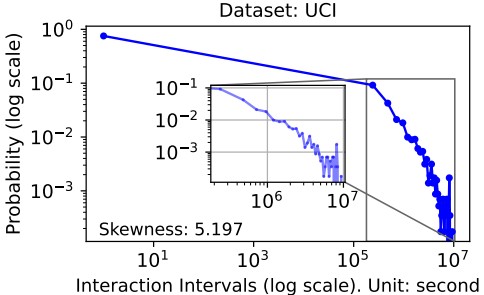

Figure 1: The interaction intervals between any pair of nodes on the UCI dataset follow a power-law distribution and have a high (positive) skewness value, meaning that the intervals are highly right-skewed.

Our key contributions can be summarized as follows:

- **First study on the heterogeneity in temporal interactions.** To the best of our knowledge, this is the first work to identify the presence of heterogeneity in temporal interactions of CTTGs and investigate its impact on the performance of temporal link prediction.

- **TAMI: a novel framework to cope with the heterogeneity in temporal interactions.** We propose TAMI, which effectively handles the heterogeneity in temporal interactions. TAMI contains two novel modules, namely LTE and LHA. LTE rescales the temporal differences using a logarithmic transformation while LHA is specifically designed to preserve historical interactions for each target node pair for link prediction. Existing temporal graph neural networks can be seamlessly and readily integrated into TAMI.

- **Extensive evaluation on open datasets**. We validate the effectiveness of TAMI on 13 classic temporal graph datasets covering different fields with comprehensive temporal scales as well as three newest ones from temporal graph benchmark (TGB). Results show that TAMI consistently and substantially improves the link prediction accuracy and training efficiency of the underlying graph neural networks, with up to 87.05% improvement in link prediction accuracy and 76.7% reduction in total training time.

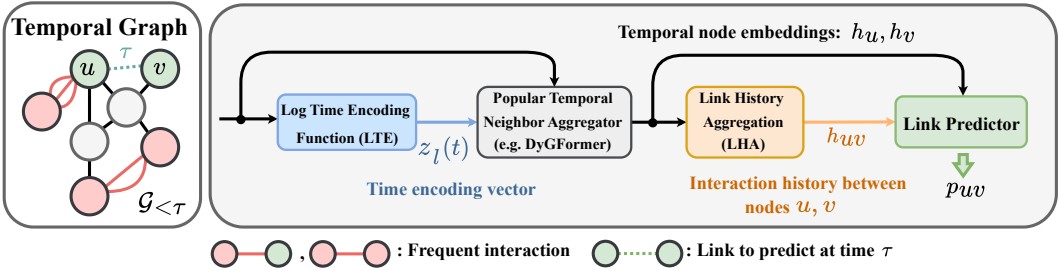

Figure 2: The TAMI framework.

## 2  Related Work

**Time Encoding Functions in Link Prediction.** Most prior methods [34, 7, 39, 31, 42, 29, 22, 6] use a common approach for time encoding, which is based on either periodic time encoding functions introduced in [33] or their simplified ones. To predict the existence of a link between two nodes at the target time, the temporal difference between the target time and the time of each historical interaction needs to be calculated. The time differences are then mapped to time encoding vectors using a periodic function such as sinusoidal function. However, this time encoding pipeline overlooks the effect of the skewness in the time differences. We observe that it takes longer than needed to train a model, and it can also degrade the performance of link prediction, especially when predicting the links for infrequently interacting node pairs. In this work, we propose LTE to effectively alleviate the skewness in the time differences via a simple yet effective logarithmic transformation, leading to better training efficiency and improved link prediction accuracy.

**Temporal Neighborhood Aggregation in Link Prediction.** In learning temporal node embeddings using information aggregation, there are typically two types of temporal graph neural networks (TGNNs), which are random walk-based TGNNs and temporal neighbor-based TGNNs. Random walk-based TGNNs [31, 10, 14] first generate multiple causal, anonymous walks for each node and then aggregate information from these walks to obtain the final node embeddings. Since frequent interactions predominantly appear as edges in the graph, it may be less likely to sample infrequent interactions in each walk. In addition, temporal neighbor-based (or graph convolution-based) TGNNs [27, 13, 22, 44, 3, 17, 34, 29, 7, 39, 25, 30, 15, 43, 16, 4] compute temporal node embeddings by aggregating node or edge features from the temporal neighbors of each target node. For example, recurrent networks or temporal point processes [27] are leveraged in [13, 3, 27] to store the states of historical interactions. DyGFormer [39] computes node embeddings by aggregating an extensive number of single-hop temporal neighbors using an attention mechanism. GraphMixer [7] proposes to use MLP-Mixer [26] and neighbor mean-pooling to compute node embeddings. However, these methods mainly rely on the recent interactions of a node to compute its embedding, which are again dominated by frequent interactions with a limited number of its neighbors. Thus, they can be less effective in predicting the links for infrequently interacting node pairs. In contrast, we propose LHA to effectively address this problem by explicitly capturing the historical interactions between each target node pair for link prediction in the modeling process.

## 3  TAMI Design

### 3.1  Problem Definition

A CTTG over a time interval $[0, T]$ is characterized by $\mathcal{G}_T = (\mathcal{V}_T, \mathcal{E}_T)$, which is a sequence of chronologically ordered interaction events between nodes up to time $T$ (inclusive), where $\mathcal{V}_T$ is the set of nodes and $\mathcal{E}_T$ is the set of temporal edges up to $T$. Here, a temporal edge $e_{uv}^t \in \mathcal{E}_T$ between two nodes $u \in \mathcal{V}_T$ and $v \in \mathcal{V}_T$ represents an interaction event between $u$ and $v$ at time $t < T$. Note that multiple edges can exist between two nodes in the temporal graph.

Given two target nodes $u$ and $v$, the target time $\tau$, and all historical interaction events in the graph up to $\tau$ (exclusive) that are characterized by $\mathcal{G}_{<\tau}$, our task is to predict how likely nodes $u$ and $v$ will interact with each other at time $\tau$, as illustrated in Figure 2. In other words, it is to estimate the probability $p_{uv}$ of having a link between $u$ and $v$ at time $\tau$. To tackle this problem, there are two common major steps involved in most TGNNs [34, 7, 39, 42, 19]. First, they compute the temporal

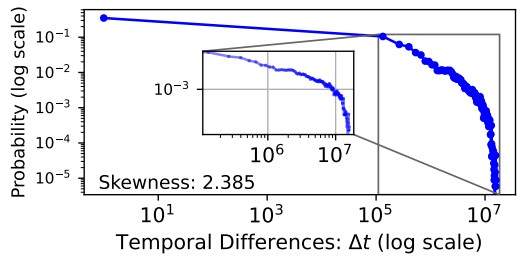 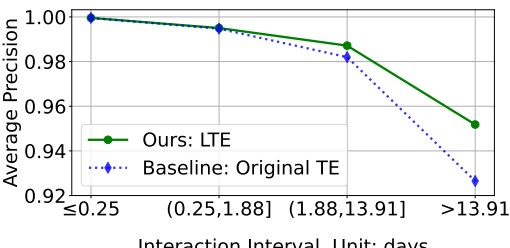

(a) The distribution of temporal difference $\Delta t$ on the UCI dataset is highly right-skewed.

(b) LTE effectively mitigates the skewness and improves the model (GraphMixer [7]) performance.

Figure 3: (a) Distribution of temporal difference $\Delta t$ and (b) model performance on the UCI dataset.

embeddings of nodes $u$ and $v$ by aggregating information from their recent neighbors who have recently had interactions with them. Second, the learned embeddings are then fed into a link predictor, e.g., an MLP with a sigmoid activation function, to estimate the probability $p_{uv}$.

### 3.2 LTE: Log Time Encoding Function

In most TGNNs [34, 7, 39, 14, 4], each historical interaction at time $t < \tau$ (or its corresponding temporal edge in $\mathcal{G}_{<\tau}$) is associated with a time encoding vector $\boldsymbol{z}(t)$, which is given by

$$\boldsymbol{z}(t) = \cos(\Delta t \times \boldsymbol{\omega}), \tag{1}$$

with $\Delta t = \tau - t$, and learnable frequency parameters $\boldsymbol{\omega} = \{\alpha^{-(i-1)/\beta}\}_{i=1}^{d_T}$, where $d_T$ is the dimension of the time encoding vector, and the values of $\alpha$ and $\beta$ are initialized as $\alpha = \beta = \sqrt{d_T}$. This time encoding function first maps $\Delta t$ to a monotonically decreasing vector $\Delta t \times \boldsymbol{\omega}$ such that the values are within $(0, \Delta t]$ and then projects them to $[-1, 1]$ using the cosine function. For the sake of clarity, we refer to this type of time encoding as original TE.

We observe that a given node can interact with various other nodes at different frequencies. Thus, their interaction intervals can vary significantly. As shown in Figure 1, the interaction intervals between nodes are highly right-skewed on the UCI dataset. Such right skewness also appears in the distribution of temporal differences $\Delta t$, as can be seen from Figure 3a. TGNNs trained on these skewed inputs using original TE learn better on the interactions that frequently appear but struggle with the ones that seldom occur. As a result, they are ineffective when making a link prediction for a pair of nodes that interact rarely or whose interaction intervals are long, as shown in Figure 3b, where we group testing node pairs for link prediction according to their average interaction intervals and report the average prediction accuracy for each group of node pairs.

To address this challenge, we propose LTE, a simple yet effective time encoding function, which rescales the value of $\Delta t$ via a logarithmic transformation so that it follows a more balanced distribution. With LTE, a large variance in $\Delta t$ no longer leads to a large discrepancy, making it easier to learn the frequency parameters $\boldsymbol{\omega}$. The time encoding function in LTE is formally defined as

$$\boldsymbol{z}_l(t) = \cos(\Delta t_l \times \boldsymbol{\omega}), \text{ with } \Delta t_l = \ln(1 + \Delta t). \tag{2}$$

To see how the logarithmic transformation mitigates the skewness in the distribution, we consider $\Delta t$ to follow a Pareto distribution, which is a power-law distribution. We have the following:

**Proposition 1.** *Suppose $\Delta t$ follows a Pareto distribution with the shape parameter $\alpha > 3$, whose skewness is always greater than 2. LTE reduces the skewness to 2.*

The proof is provided in Appendix A.

In practice, $\Delta t$ may not strictly follow a Pareto distribution. Nonetheless, LTE can still effectively mitigate its skewness, thereby improving the performance of the underlying model. As shown in Figure 3b, LTE improves the model performance by reducing the skewness of $\Delta t$ on the UCI dataset from 2.385 to $-1.14$. Please refer to Table 11 in Appendix B.5 for the values of skewness with and without LTE on different datasets.

Once the time encoding vector is obtained, the next step in TAMI is to compute the temporal node embeddings. For this purpose, any existing TGNN [27, 13, 22, 34, 29, 7, 39, 31] can be adopted. In temporal neighbor-based methods [39, 7, 34, 13, 22], the temporal embedding $\boldsymbol{h}_u \in \mathbb{R}^d$ of node $u$ is generally computed based on the $m$ most recent interactions of $u$ with its temporal neighbors. Specifically, let $\mathcal{N}_u$ be the set of the $m$ recent neighbors of $u$, $\boldsymbol{x}_j$ be the initial embedding of node $j$, and $\boldsymbol{x}_{uj}$ be the initial embedding of temporal edge $e_{uj}$, and $\boldsymbol{z}_l(t_j)$ be the time encoding vector for the interaction event with $j$ at time $t_j$. Then, the embedding $\boldsymbol{h}_u$ is obtained as follows:

$$\boldsymbol{h}_u = \text{AGGREGATE}\left(\{[\boldsymbol{x}_j; \boldsymbol{x}_{uj}; \boldsymbol{z}_l(t_j)]\}_{j \in \mathcal{N}_u}\right), \tag{3}$$

where $\text{AGGREGATE}(\cdot)$ is an aggregation function and $[;]$ is the concatenation operation. Similarly, random-walk based methods [31, 10] obtain the embedding $\boldsymbol{h}_u$ by aggregating the information from nodes that appear in random walks starting from $u$. Both classes of methods can be readily integrated into TAMI, as shall be demonstrated in Section 4.5.

## 3.3 LHA: Link History Aggregation

As mentioned above, it is common practice that the temporal embedding of a node $u$ is based only on its most recent $m$ interactions or the recent ones that appear in random walks from $u$. However, this can be problematic for predicting a link between two nodes $u$ and $v$, especially when they do not appear in their mutual nearest neighbors. In that case, their historical interactions are forgotten in updating their temporal embeddings, thereby degrading the link prediction accuracy. See Figure 4 for an illustration. We empirically observe that this is indeed the case, as shown in Figure 5. There is a non-negligible (possibly significant) portion of node pairs that could have forgotten their mutual interaction history for temporal link prediction, and their link prediction accuracy is the worst.

To resolve this problem, we propose a novel light-weight module called link history aggregation (LHA). Its core idea is to preserve the most recent $k$ interactions for a target node pair and leverage this historical information, along with their temporal node embeddings, to predict a link between the target node pair. Let $e_{uv}^{t_1}, e_{uv}^{t_2}, \ldots, e_{uv}^{t_k}$ denote the most recent $k$ interactions between nodes $u$ and $v$ that happen at times $t_1, t_2, \ldots, t_k$ before time $\tau$, respectively, where $t_k < t_{k-1} < \cdots < t_1 < \tau$. The $i$-th historical interaction $e_{uv}^{t_i}$ is associated with a $d_r$-dimensional historical edge embedding $\boldsymbol{r}_{uv}^{t_i} \in \mathbb{R}^{d_r}$, which encodes the information of the interaction. We use $M_{uv}(\tau) = \{\boldsymbol{r}_{uv}^{t_1}, \boldsymbol{r}_{uv}^{t_2}, ..., \boldsymbol{r}_{uv}^{t_k}\}$ to indicate the historical

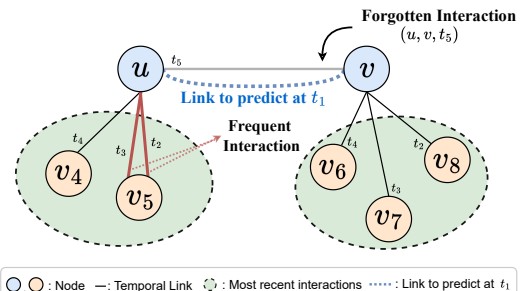

Figure 4: When predicting the future link between $u$ and $v$ at $t_1$, their historical interaction at $t_5$ is no longer retained in their latest node embeddings.

edge embeddings of the most recent $k$ interactions before time $\tau$. Then, the edge embedding vector $\boldsymbol{r}_{uv}^\tau$ for link prediction at time $\tau$ is defined as a weighted sum of the current node embeddings and the previous edge embedding as:

$$\boldsymbol{r}_{uv}^\tau = \gamma \times \boldsymbol{c}_{uv} + (1 - \gamma) \times \boldsymbol{r}_{uv}^{t_1}, \tag{4}$$

where $\boldsymbol{c}_{uv} = \text{MLP}([\boldsymbol{h}_u\,;\,\boldsymbol{h}_v]) \in \mathbb{R}^{d_r}$ encodes the current states of nodes $u$ and $v$, with $\boldsymbol{h}_u$ and $\boldsymbol{h}_v$ being the temporal node embeddings of nodes $u$ and $v$, respectively, and $\boldsymbol{r}_{uv}^{t_1}$ is the most recent edge embedding in $M_{uv}(\tau)$. Here the hyperparameter $\gamma \in [0, 1]$ controls the 'forgetting' rate of historical interactions. For example, if $\gamma = 1$, the entire interaction history is discarded in computing $\boldsymbol{r}_{uv}^\tau$. To bootstrap the link prediction between $u$ and $v$ which do not have interaction history, i.e. $M_{uv}(\tau) = \emptyset$, we set $\boldsymbol{r}_{uv}^{t_1} = \boldsymbol{0}$.

To predict a link between nodes $u$ and $v$ at time $\tau$ as the last step of TAMI, we first aggregate all the historical link embeddings in $M_{uv}(\tau)$ into a single vector $\boldsymbol{h}_{uv}$, which summarizes the most recent $k$ interaction histories between $u$ and $v$. This can be written as

$$\boldsymbol{h}_{uv} = \text{AGGREGATE}\left(\boldsymbol{r}_{uv}^{t_1}, \boldsymbol{r}_{uv}^{t_2}, ..., \boldsymbol{r}_{uv}^{t_k}\right) \in \mathbb{R}^{d_r}, \tag{5}$$

where $\text{AGGREGATE}(\cdot)$ denotes an aggregation function. While various options can be considered (such as sum, mean, or attention), we employ the **most-recent** aggregator. It utilizes the edge

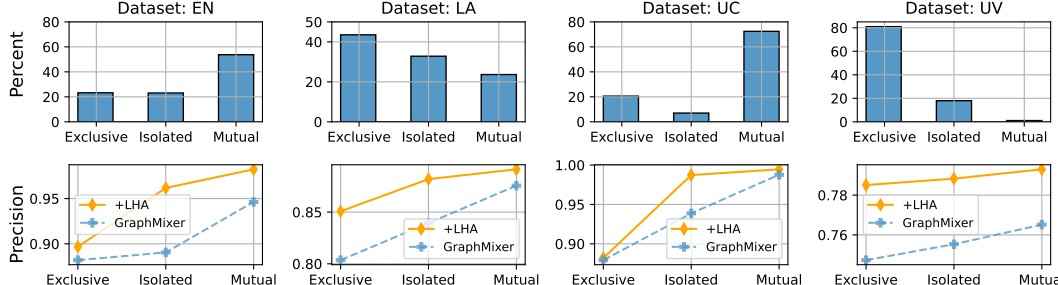

Figure 5: The performance of GraphMixer and its improvement with LHA. "Exclusive" means that neither of two nodes appears in the other's $m$ recent interactions. "Isolated" indicates that only one node appears in the other's $m$ recent interactions. "Mutual" means that both nodes appear.

embedding of the most recent historical interaction to update $\boldsymbol{h}_{uv}$, i.e., $\boldsymbol{h}_{uv} = \boldsymbol{r}_{uv}^{t_1}$. Finally, the link probability $p_{uv}$ is computed using an MLP with the sigmoid activation function, i.e.,

$$p_{uv} = \text{MLP}\big([\boldsymbol{h}_u\,;\,\boldsymbol{h}_v\,;\,\boldsymbol{h}_{uv}]\big). \tag{6}$$

Whenever an interaction between nodes $u$ and $v$ occurs, their corresponding history set $M_{uv}(\cdot)$ needs to be updated. Suppose $u$ and $v$ are connected at time $\tau$. A new edge embedding vector $\boldsymbol{r}_{uv}^{\tau}$, computed using Equation (4), is added to $M_{uv}(\tau)$ for the next link prediction. If the original size of $M_{uv}(\tau)$ is equal to $k$, we first remove the oldest historical edge embedding from $M_{uv}(\tau)$ and then insert $\boldsymbol{r}_{uv}^{\tau}$. In other words, we keep the most recent $k$ interactions only.

**Remarks.** As shown in Figure 5 and shall be demonstrated in the subsequent section, LHA improves the performance of the underlying TGNN, with up to 25.33% improvement in link prediction accuracy. LHA also has high efficiency in terms of GPU memory usage. Note that $\boldsymbol{h}_{uv}$ is the only additional embedding used in computing the link probability $p_{uv}$, and we use the *most-recent* aggregator to obtain $\boldsymbol{h}_{uv} = \boldsymbol{r}_{uv}^{t_1}$. Also, the historical edge embedding $\boldsymbol{r}_{uv}^{t_1}$ is updated based on its previous one, as in Equation (4). Thus, letting $N$ be the number of target node pairs for prediction in the dataset, the total space complexity of LHA is $O(Nd_r)$, where $d_r$ is the dimension of each historical edge embedding. Since mini-batch computation can be employed, it is unnecessary to load the edge embeddings for all target node pairs in GPU memory. Instead, they can be stored in CPU's system memory and loaded dynamically as required. Therefore, the additional GPU memory overhead introduced by LHA is $O(bd_r)$, where $b$ denotes the mini-batch size, and $b \ll N$. Furthermore, with a slight increase in complexity, LHA also speeds up the convergence of the underlying model in training, with up to 76.7% reduction in total training time, as shall be demonstrated shortly.

## 4 Experiments

### 4.1 Experimental Settings

We conduct experiments on 13 classic open datasets [21]: Can. Parl. (**CP**), Contact (**CO**), Enron (**EN**), Flights (**FL**), Lastfm (**LA**), Mooc (**MO**), Reddit (**RE**), Social Evo (**SE**), Uci (**UC**), UN Trade (**UT**), UN Vote (**UV**), US Legis (**US**), and Wikipedia (**WK**). These datasets cover various domains and their details are provided in Section B.1. We integrate two state-of-the-art TGNNs, namely GraphMixer [7] and DyGFormer [39], into our TAMI framework. More specifically, we use their temporal neighbor aggregation function in the Temporal Neighbor Aggregator module of TAMI. We compare the performance with their vanilla counterparts, as well as seven other state-of-the-art TGNNs for CTTGs, including JODIE [13], DyRep [27], TGAT [34], TGN [22], CAWN [31], Edgebank [21], and TCL [29]. Descriptions of the baselines are provided in Section B.2.

We evaluate the link prediction performance of TGNNs under two settings: (1) the transductive setting where future links are predicted between nodes observed during training, and (2) the inductive setting where predictions are made for nodes unseen during training. Following [21, 39], we chronologically split each dataset into 70%/15%/15% for training/validation/testing, and adopt the average precision (AP) score as the evaluation metric. We present the implementation details in Section B.3 and baseline configurations in Section B.4. Unless otherwise specified, we below present the major results for the transductive setting only and report additional results for the transductive and inductive settings in Section C and Section D, respectively. For the generation of negative links, we follow

Table 1: AP for transductive link prediction under three different negative sampling strategies (NSS). Imp. (%) denotes the percentage of *improvement*. The first and the second best performers are marked in **bold** and underlined, respectively. Standard deviations over five runs are reported in Table 21.

| NSS | Methods | CP | CO | EN | FL | LA | MO | RE | SE | UC | UT | UV | US | WK |
|---|---|---|---|---|---|---|---|---|---|---|---|---|---|---|
| rnd | JODIE | 69.26 | 95.31 | 84.77 | 95.60 | 70.85 | 80.23 | 98.31 | 89.89 | 89.43 | 64.94 | 63.91 | 75.05 | 96.50 |
| | DyRep | 66.54 | 95.98 | 82.38 | 95.29 | 71.92 | 81.97 | 98.22 | 88.87 | 65.14 | 63.21 | 62.81 | 75.34 | 94.86 |
| | TGAT | 70.73 | 96.28 | 71.12 | 94.03 | 73.42 | 85.84 | 98.52 | 93.16 | 79.63 | 61.47 | 52.21 | 68.52 | 96.94 |
| | TGN | 70.88 | 96.89 | 86.53 | 97.95 | 77.07 | **89.15** | 98.63 | 93.57 | 92.34 | 65.03 | **65.72** | **75.99** | 98.45 |
| | CAWN | 69.82 | 90.26 | 89.56 | 98.51 | 86.99 | 80.15 | 99.11 | 84.96 | 95.18 | 65.39 | 52.84 | 70.58 | 98.76 |
| | EdgeBank | 64.55 | 92.58 | 83.53 | 89.35 | 79.29 | 57.97 | 94.86 | 74.95 | 76.20 | 60.41 | 58.49 | 58.39 | 90.37 |
| | TCL | 68.67 | 92.44 | 79.70 | 91.23 | 67.27 | 82.38 | 97.53 | 93.13 | 89.57 | 62.21 | 51.90 | 69.59 | 96.47 |
| | GraphMixer | 75.90 | 91.94 | 82.26 | 90.98 | 75.56 | 82.83 | 97.33 | 93.34 | 93.38 | 62.61 | 52.20 | 71.55 | 97.23 |
| | DyGFormer | 97.91 | 98.31 | 92.46 | 98.92 | 93.01 | 87.66 | 99.22 | 94.66 | 95.66 | 65.07 | 55.88 | 70.44 | 99.02 |
| | **with TAMI** | | | | | | | | | | | | | |
| | GraphMixer | 78.38 | 95.26 | 90.97 | 96.75 | 88.13 | 83.53 | 98.84 | 93.41 | 96.20 | 62.98 | 57.74 | 71.57 | 98.89 |
| | Imp. (%) | 3.27% | 3.61% | 10.59% | 6.34% | 16.64% | 0.85% | 1.56% | 0.07% | 3.02% | 0.59% | 10.61% | 0.03% | 1.71% |
| | DyGFormer | **98.67** | **98.70** | **92.66** | **98.94** | **94.03** | 88.49 | **99.29** | **94.74** | **96.72** | **66.39** | 56.02 | 71.40 | **99.25** |
| | Imp. (%) | 0.78% | 0.40% | 0.22% | 0.02% | 1.10% | 0.95% | 0.07% | 0.08% | 1.11% | 2.03% | 0.25% | 1.36% | 0.23% |
| hist | GraphMixer | 74.34 | 93.29 | 77.98 | 71.47 | 72.47 | 77.77 | 78.44 | 94.93 | 84.11 | 57.05 | 51.20 | 81.65 | 90.90 |
| | DyGFormer | 97.00 | 97.57 | 75.63 | 66.59 | 81.57 | 85.85 | 81.57 | 97.38 | 82.17 | 64.41 | 60.84 | 85.30 | 82.23 |
| | **with TAMI** | | | | | | | | | | | | | |
| | GraphMixer | 78.81 | 93.30 | 81.68 | 73.01 | 80.23 | 83.61 | 82.56 | 96.80 | 87.69 | 69.74 | 70.90 | 84.56 | 90.97 |
| | Imp. (%) | 6.01% | 0.01% | 4.74% | 2.15% | 10.71% | 7.51% | 5.25% | 1.97% | 4.26% | 22.24% | 38.48% | 3.56% | 0.08% |
| | DyGFormer | 98.96 | 97.72 | 81.02 | 67.77 | 83.40 | 86.26 | 85.18 | 97.56 | 85.89 | 65.16 | 81.72 | 86.10 | 82.38 |
| | Imp. (%) | 2.02% | 0.15% | 7.13% | 1.77% | 2.24% | 0.48% | 4.43% | 0.18% | 4.53% | 1.16% | 34.32% | 0.94% | 0.18% |
| ind | GraphMixer | 69.48 | 90.87 | 75.01 | 74.87 | 68.12 | 74.26 | 85.26 | 94.72 | 80.10 | 60.15 | 51.60 | 79.63 | 88.59 |
| | DyGFormer | 95.44 | 94.75 | 77.41 | 70.92 | 73.97 | 81.24 | 91.11 | 97.68 | 72.25 | 55.79 | 51.91 | 81.25 | 78.29 |
| | **with TAMI** | | | | | | | | | | | | | |
| | GraphMixer | 70.94 | 96.12 | 88.95 | 93.64 | 91.06 | 79.82 | 96.19 | 96.09 | 84.12 | 87.73 | 79.53 | 83.31 | 93.89 |
| | Imp. (%) | 2.10% | 5.78% | 18.58% | 25.07% | 33.68% | 7.49% | 12.82% | 1.45% | 5.02% | 45.85% | 54.13% | 4.62% | 5.98% |
| | DyGFormer | 97.25 | 98.47 | 86.23 | 75.55 | 74.03 | 92.39 | 94.37 | 97.76 | 80.13 | 68.01 | 78.19 | 81.31 | 78.96 |
| | Imp. (%) | 1.90% | 3.93% | 11.39% | 6.53% | 0.08% | 13.72% | 3.58% | 0.08% | 10.91% | 21.90% | 50.63% | 0.07% | 0.86% |

the negative sampling strategies in [21], where random negative sampling is used for training and random, historical, and inductive negative samplings are applied during evaluation, with random negative sampling as the default choice.

## 4.2 Main Results

**Performance under different negative sampling strategies**. As shown in Table 1, TAMI substantially improves the performance of the integrated models under all three negative sampling strategies. In particular, for the random negative sampling strategy (rnd), the performance of both GraphMixer and DyGFormer improves on all 13 datasets, with improvement up to 16.64%. This is because LTE balances the skewness in temporal differences during the time encoding process, especially for nodes that interact rarely or whose interaction intervals are long, while LHA prevents historical interactions between the target node pair from being forgotten in predicting their future links, improving the link prediction accuracy. In addition, we show in Section D.1 that TAMI remains effective in the inductive setting, consistently improving the performance of the integrated models. These results validate the effectiveness and versatility of the proposed TAMI framework.

We also present the results of TAMI under the historical (hist) and the inductive (ind) negative sampling strategies in Table 1. As shown in Table 1, TAMI consistently improves TGNNs under both negative sampling strategies, achieving improvements of up to 38.48% and 54.13% for the historical and inductive negative sampling strategies, respectively. This is because LHA maintains the interaction histories of node pairs as the test stage progresses, allowing the underlying models to better capture historical interaction patterns under historical sampling and inductive sampling, more accurately predicting future connections. Please refer to Table 12 for full results.

Table 2: Test mean reciprocal rank (MRR) scores on TGB datasets. TGB leaderboard is publicly available.

| Datasets | DyGFormer | w/ TAMI | Imp (%) |
|---|---|---|---|
| tgbl-wiki | 0.798 (rank 1) | **0.815** (rank 1) | 2.13% |
| tgbl-review | 0.224 (rank 6) | **0.419** (rank 2) | 87.05% |
| tgbl-coin | 0.752 (rank 2) | **0.794** (rank 1) | 5.59% |

Table 3: AP for transductive link prediction.

| Methods | CP | CO | EN | FL | LA | MO | RE | SE | UC | UT | UV | US | WK |
|---|---|---|---|---|---|---|---|---|---|---|---|---|---|
| GraphMixer | 75.90 | 91.94 | 82.26 | 90.98 | 75.56 | 82.83 | 97.33 | 93.34 | 93.38 | 62.61 | 52.20 | 71.55 | 97.23 |
| w/ LTE | 78.07 | 92.22 | 82.76 | 90.99 | 75.21 | 83.09 | 97.35 | 92.64 | 94.98 | 62.65 | 52.21 | 70.88 | 97.28 |
| Imp. (%) | 2.86% | 0.30% | 0.61% | 0.01% | -0.46% | 0.31% | 0.02% | -0.75% | 1.71% | 0.06% | 0.02% | -0.94% | 0.05% |
| w/ LHA | 75.93 | 95.15 | 89.88 | 96.72 | 88.15 | 83.36 | 98.81 | 93.71 | 94.90 | 62.83 | 57.57 | 71.61 | 98.85 |
| Imp. (%) | 0.04% | 3.49% | 9.26% | 6.31% | 16.66% | 0.64% | 1.52% | 0.40% | 1.63% | 0.35% | 10.30% | 0.08% | 1.67% |
| w/ TAMI | 78.38 | 95.26 | 90.97 | 96.75 | 88.13 | 83.53 | 98.84 | 93.41 | 96.20 | 62.98 | 57.74 | 71.57 | 98.89 |
| Imp. (%) | 3.27% | 3.61% | 10.59% | 6.34% | 16.64% | 0.85% | 1.55% | 0.07% | 3.02% | 0.59% | 10.61% | 0.03% | 1.71% |
| DyGFormer | 97.91 | 98.31 | 92.46 | 98.92 | 93.01 | 87.66 | 99.22 | 94.66 | 95.66 | 65.07 | 55.88 | 70.44 | 99.02 |
| w/ LTE | 98.73 | 98.36 | 92.61 | 98.93 | 93.92 | 88.59 | 99.27 | 94.74 | 96.68 | 67.24 | 56.37 | 70.98 | 99.23 |
| Imp. (%) | 0.84% | 0.05% | 0.16% | 0.02% | 0.98% | 1.06% | 0.06% | 0.08% | 1.06% | 3.34% | 0.88% | 0.77% | 0.21% |
| w/ LHA | 97.57 | 98.65 | 92.59 | 98.92 | 93.44 | 87.55 | 99.23 | 94.72 | 96.00 | 64.53 | 55.99 | 71.23 | 99.07 |
| Imp. (%) | -0.35% | 0.34% | 0.14% | 0.00% | 0.47% | -0.13% | 0.02% | 0.07% | 0.36% | -0.83% | 0.20% | 1.12% | 0.05% |
| w/ TAMI | 98.67 | 98.70 | 92.66 | 98.94 | 94.03 | 88.49 | 99.29 | 94.74 | 96.72 | 66.39 | 56.02 | 71.40 | 99.25 |
| Imp. (%) | 0.78% | 0.40% | 0.22% | 0.03% | 1.10% | 0.95% | 0.07% | 0.08% | 1.11% | 2.03% | 0.25% | 1.36% | 0.23% |

**Performance on three datasets from temporal graph benchmark (TGB).** We further evaluate TAMI on the *tgbl-wiki*, *tgbl-review*, and *tgbl-coin* datasets from TGB [9]. As shown in Table 2, incorporating DyGFormer into the proposed TAMI framework significantly improves its performance, making it the top-performing model on the *tgbl-wiki* and *tgbl-coin* datasets, and the second-best model on the *tgbl-review* dataset. This is because LTE is robust to different levels of skewness in datasets (see Section C.3) and LHA improves prediction accuracy whenever the interaction history between the target node pair are informative in predicting their future link. Detailed dataset statistics and experimental settings are provided in Section E.

## 4.3 Ablation Study

We conduct experiments to examine the effectiveness of the proposed LTE and LHA modules. Table 3 presents the test performance of TAMI and its two variants: **w/ LTE**, where we replace the original TE in TGNNs with the proposed LTE and keep the rest unchanged; **w/ LHA**, where we integrate the LHA module into TGNNs and keep the rest unchanged. As shown from the results, both LTE and LHA improve the underlying TGNN performance when integrated individually. The performance further boosts when they are combined together. This indicates that our designed LTE and LHA are highly effective and versatile across datasets with various domains and temporal scales. We demonstrate in Section D.2 that LTE and LHA are also effective in the inductive setting, consistently enhancing the performance of integrated TGNNs.

## 4.4 Robustness to the Increase of Negative Links

In this experiment, we evaluate the robustness of TAMI against an increasing number of negative links per positive link. A negative link refers to a pair of nodes that are not currently connected and are used as negative samples in the link prediction process. Ideally, a link prediction model should assign a higher connection probability to positive links and a probability close to zero to negative links. *The more negative links, the more difficult the task is.* In the default setting, the number of negative links is set to 1 per positive link. We run the experiments on four representative datasets (EN, LA, UC, and UV), and report the results in Table 4. "NEG=50" denotes that each positive link is evaluated against 50 negative links when computing its connection probability.

Table 4: AP of methods under various numbers of negative links during testing. **NEG=50** indicates that each positive link is evaluated against 50 negative links in the AP computation.

| Method | EN | LA | UC | UV | | Method | EN | LA | UC | UV |
|---|---|---|---|---|---|---|---|---|---|---|
| **NEG = 1** | | | | | | | | | | |
| GraphMixer | 82.26 | 75.56 | 93.38 | 52.20 | | DyGFormer | 92.46 | 93.01 | 95.66 | 55.88 |
| w/ TAMI | 90.97 | 88.13 | 96.20 | 57.74 | | w/ TAMI | 92.66 | 94.03 | 96.72 | 56.02 |
| Imp (%) | 10.59% | 16.64% | 3.02% | 10.61% | | Imp (%) | 0.22% | 1.10% | 1.11% | 0.25% |
| **NEG = 5** | | | | | | | | | | |
| GraphMixer | 53.42 | 47.06 | 82.05 | 18.17 | | DyGFormer | 75.81 | 77.77 | 88.77 | 20.86 |
| w/ TAMI | 70.36 | 65.05 | 88.40 | 21.62 | | w/ TAMI | 76.37 | 80.58 | 90.76 | 20.97 |
| Imp (%) | 31.72% | 38.24% | 7.73% | 18.99% | | Imp (%) | 0.74% | 3.61% | 2.24% | 0.55% |
| **NEG = 25** | | | | | | | | | | |
| GraphMixer | 22.77 | 23.77 | 63.54 | 4.36 | | DyGFormer | 45.15 | 51.58 | 78.96 | 5.13 |
| w/ TAMI | 36.57 | 34.97 | 73.26 | 5.32 | | w/ TAMI | 46.39 | 56.61 | 81.74 | 5.28 |
| Imp (%) | 60.61% | 47.12% | 15.30% | 22.04% | | Imp (%) | 2.74% | 9.74% | 3.52% | 2.92% |
| **NEG = 50** | | | | | | | | | | |
| GraphMixer | 14.09 | 16.80 | 54.14 | 2.25 | | DyGFormer | 31.06 | 35.64 | 73.47 | 2.65 |
| w/ TAMI | 23.43 | 24.66 | 63.74 | 2.74 | | w/ TAMI | 32.43 | 44.76 | 76.59 | 2.72 |
| Imp (%) | 66.29% | 46.76% | 17.74% | 21.78% | | Imp (%) | 4.41% | 25.59% | 4.25% | 2.64% |

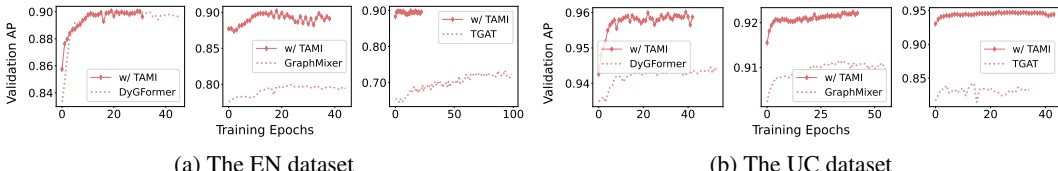

|   (a) The EN dataset   |   (b) The UC dataset   |

Figure 6: Validation AP vs. training epochs on the (a) EN and (b) UC datasets. TAMI enables TGNNs to achieve higher validation average precision with fewer training epochs.

As shown in Table 4, integrating GraphMixer and DyGFormer into TAMI consistently enhances their performance across different numbers of negative links per positive link. Furthermore, the improvement ratio steadily increases as the number of negative links grows. For example, on the EN dataset, the improvement ratio for GraphMixer increases from 10.59% to 66.29% as the number of negative links per positive link increases from 1 to 50. This is because leveraging the interaction histories stored in LHA between target node pairs helps to avoid predicting negative links as positive ones. These results suggest that the proposed TAMI framework effectively improves TGNN performance in scenarios where the ratio of negative links to positive links is high, reflecting conditions that are more practical for sparse CTTGs.

## 4.5 Adaptivity to Different TGNN Architectures

We below show the adaptivity of TAMI to different TGNN architectures. Specifically, we integrate a random walk-based TGNN, i.e. CAWN [31], and three temporal neighbor-based TGNNs, i.e., TGAT [34], TGN [22], and JODIE [13], into TAMI. Since JODIE does not utilize the time encoding function, we report its performance with LHA integrated. As shown in Table 5,

Table 5: AP Improvement (Imp.%) of various TGNNs when integrated into TAMI. The format is of AP (+Imp.%). Please refer to Table 13 for full results.

| Method (w/ TAMI) | EN | LA | UC | UV |
|---|---|---|---|---|
| CAWN | 91.23 (+3.21%) | 91.02 (+4.64%) | 96.69 (+1.81%) | 57.49 (+8.72%) |
| TGAT | 91.37 (+25.35%) | 91.60 (+24.85%) | 96.36 (+21.53%) | 60.03 (+12.98%) |
| TGN | 92.34 (+6.03%) | 92.83 (+22.58%) | 95.36 (+3.80%) | 67.80 (+3.22%) |
| JODIE | 90.62 (+6.88% ) | 87.95 (+25.33%) | 92.44 (+3.68%) | 65.57 (+3.22%) |

TAMI consistently improves their performance, indicating the effectiveness of TAMI for different TGNN architectures. Additionally, we show in Table 13 that applying LTE and LHA individually also boosts the performance of the underlying TGNNs.

## 4.6 Improved Training Efficiency

We observe that TAMI can speed up the training of underlying TGNNs. To show this, we plot the validation AP of TGNNs during training in Figure 6. On the EN and UC datasets, TGNNs within the TAMI framework achieve higher validation AP scores with *much fewer* training epochs, compared to their vanilla counterparts. In other words, the underlying models converge faster than their vanilla versions. These findings suggest that TAMI can speed up the convergence of TGNNs while yielding improved validation AP scores. This is because LTE facilitates the learning of frequency parameters in the time encoding function by balancing input temporal differences, while LHA enables the underlying model to more effectively capture the patterns of historical interactions between target node pairs.

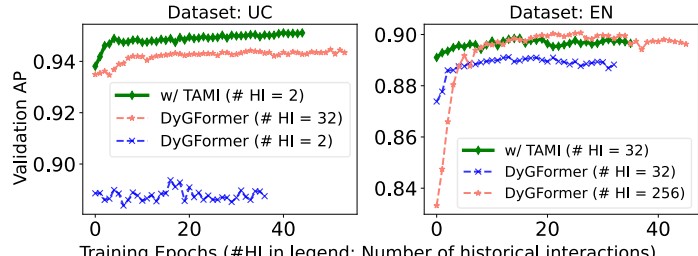

| Metric
Method | GPU Memory
Usage (MiB) | | Training Time
Per Epoch (Second) | |
|---|---|---|---|---|
| | UC | EN | UC | EN |
| DyGFormer (32, 256) | 2429 | 3303 | 37 | 81 |
| DyGFormer (2, 32) w/ TAMI | 1359 | 1545 | 22 | 57 |
| Imp. (%) | 44.05% | 53.22% | 40.54% | 29.63% |

Figure 7: DyGFormer integrated with TAMI can compute temporal node embeddings using significantly fewer historical interactions while achieving better performance.

In addition, we demonstrate that DyGFormer, when integrated into our TAMI framework, can compute temporal node embeddings using *significantly fewer* historical interactions while achieving equal or even better performance. As shown in Figure 7, when integrated into TAMI, DyGFormer on the UC and EN datasets requires only 2 and 32 historical interactions for attention calculation, respectively, outperforming the vanilla counterpart that uses $16\times$ and $8\times$ more historical interactions. The reduced number of historical interactions results in lower GPU memory usage (up to 53.22%). Moreover, on both datasets, DyGFormer under the TAMI framework converges much faster than the vanilla version, with fewer training epochs and up to 40.54% reduction in training time per epoch.

## 5 Conclusion

In this paper, we observed the presence of heterogeneity in temporal interactions of CTTGs and proposed a novel framework TAMI to address the challenges therein. TAMI has two main modules, namely LTE and LHA. LTE balances the skewness in the time differences via a simple yet effective logarithmic transformation, and LHA prevents the historical interactions for each target node pair from being forgotten. Existing temporal graph neural networks can be seamlessly and readily integrated into TAMI. Extensive experiments on 13 classic open datasets and three TGB datasets show that TAMI substantially improves the link prediction accuracy of the underlying models as well as their training efficiency, in both transductive and inductive settings.

## Acknowledgments

We thank the anonymous reviewers for their constructive feedback. This work was supported in part by the Guangdong Provincial Key Laboratory of IRADS (2022B1212010006), the Guangdong Higher Education Upgrading Plan (2021-2025), and the Guangdong and Hong Kong Universities "1+1+1" Joint Research Collaboration Scheme (No. 2025A0505000004 and No. 2025A0505000012). This work was also supported in part by the National Science Foundation under Grant No. IIS-2209921, and the International Energy Joint R&D Program of the Korea Institute of Energy Technology Evaluation and Planning (KETEP), granted financial resource from the Ministry of Trade, Industry & Energy, Republic of Korea (No. 20228530050030).

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

## Technical Appendix and Supplementary Material

In the Appendix, we provide additional supplementary material to the main paper. The structure is as follows:

1. Section A outlines the proof of Proposition 1.
2. Section B outlines the experimental settings in detail.
3. Section C presents additional results for link prediction in the transductive setting.
4. Section D shows the results in the inductive setting.
5. Section E outlines the experimental settings on TGB datasets.
6. Section F presents the standard deviations of the results in the main text tables.

## A Proof of Proposition 1

Suppose $\Delta t$ follows a Pareto distribution with the shape parameter $\alpha > 3$, whose skewness is always greater than 2. Note that if the shape parameter is $0 < \alpha \leq 2$, the variance is infinite, so the skewness is undefined. Also, if the shape parameter is $2 < \alpha \leq 3$, its third moment is infinite. Thus, we focus on the case of $\alpha > 3$. Then, for any value of $\alpha > 3$, we show that LTE always reduces the skewness of $\Delta t$ to 2.

*Proof.* Let $T$ be a random variable that represents the temporal difference $\Delta t$ and follows a Pareto distribution. Then, $T + 1$ also follows a (shifted) Pareto distribution. Letting $X = T + 1$, we can write the probability density function (PDF) of $X$ as

$$f_X(x) = \frac{\alpha x_{\min}^{\alpha}}{x^{\alpha+1}}, \quad x \geq x_{\min},$$

where $\alpha > 3$ is the shape parameter, and $x_{\min} > 1$ is the scale parameter.

First, we show that the skewness $\Gamma$ of $X$ is always greater than 2. We write the skewness $\Gamma$ of $X$ as a function of $\alpha$, i.e.,

$$\Gamma = g(\alpha) = \frac{2(1+\alpha)}{\alpha - 3} \sqrt{1 - \frac{2}{\alpha}} \tag{7}$$

It is then straightforward to see that $g(\alpha)$ is monotonically decreasing, with $g(\alpha) \to 2$ as $\alpha \to \infty$. To show that $g(\alpha)$ is monotonically decreasing, we can show that $\ln g(\alpha)$ exists and $\ln g(\alpha)$ is monotonically decreasing. We first take the natural logarithm on both sides of (7) and differentiate $\ln g(\alpha)$ with respect to $\alpha$. We can then show that $\frac{g'(\alpha)}{g(\alpha)} < 0$. In addition, we can easily see from (7) that $g(\alpha) \to 2$ as $\alpha \to \infty$. Therefore, the skewness $\Gamma$ of $X$ is always greater than 2.

Next, for any value of $\alpha > 3$, we show that LTE always reduces the skewness $\Gamma$ to 2. Recall that LTE transforms $X$ via a logarithmic function $Y = \ln(X)$. By using the change-of-variables formula for probability distributions, we can obtain the PDF of $Y$ as follows:

$$f_Y(y) = f_X(x) \cdot \left| \frac{dx}{dy} \right| = \alpha x_{\min}^{\alpha} e^{-\alpha y}.$$

Letting $\mu = \ln(x_{\min}) > 0$, we have

$$f_Y(y) = \alpha e^{-\alpha(y-\mu)}, \quad y \geq \mu,$$

which is a shifted exponential distribution with the rate parameter $\alpha > 3$. Since the skewness of a (shifted) exponential distribution is 2 regardless of the value of the rate parameter $\alpha$, LTE always reduces the skewness to 2. □

## B Experimental Settings

### B.1 Descriptions of Datasets

We use 13 datasets collected by [21] in our experiments.

1. **Can. Parl. (CP)** is a dynamic political network that captures the interactions among Canadian Members of Parliament (MPs) from 2006 to 2019. Each node represents an MP from an electoral district, and an edge is established when two MPs cast a "yes" vote on the same bill. The weight of each edge reflects the annual frequency with which one MP votes "yes" alongside another.

2. **Contact (CO)** describes how the physical proximity evolves among about 700 university students over a month. Each student has a unique ID and edges between students denote that they are within close proximity to each other. Each edge is assigned a weight that reflects the physical proximity between students.

3. **Enron (EN)** is an email correspondence dataset that records the emails exchanged among employees of the ENRON energy company over three years.

4. **Flights (FL)** is a dynamic flight network illustrating the development of air traffic during the COVID-19 pandemic. Nodes represent airports and the tracked flights are denoted as edges. The edge weights reflect the number of flights between two airports in a day.

5. **LastFM (LA)** records the interaction between users and songs. Users and songs are nodes and edges between them represent a user-listens-to-song relation. The dataset contains no attributes.

6. **Mooc (MO)** is a dataset that captures students' interactions with online course materials. Each edge represents a student accessing a content unit and is associated with a 4-dimensional feature vector.

7. **Reddit (RE)** comprises user posts submitted to subreddits over one month. Users and subreddits are nodes, while timestamped posting requests form the edges. Edge features are LIWC-feature vectors [18] of edit texts.

8. **Social Evo. (SE)** is a mobile phone proximity network that tracks the daily interactions of an undergraduate dormitory over eight months. Each edge is associated with a 2-dimensional feature vector.

9. **UCI (UC)** is an unattributed online communication network among university students. Nodes are university students and edges are messages posted by students.

10. **UN Trade (UT)** is a food and agriculture trading graph between 181 nations for more than 30 years. The edge weights indicate the total sum of normalized agriculture import or export values between two countries.

11. **UN Vote (UV)** captures roll-call voting behavior in the United Nations General Assembly. Each time two nations vote a "yes" on the same item, the edge weight between them is incremented by one.

12. **US Legis. (US)** is a senate co-sponsorship graph that captures the social dynamics among US legislators. The edge weights specify the number of times two congresspersons have co-sponsored a bill in a given Congress.

13. **Wikipedia (WK)** records the edits on Wikipedia pages over a month. Editors and Wiki pages are modeled as nodes, and posting requests are timestamped edges. Edge features are 172-dimensional LIWC feature vectors [18].

We present the dataset statistics in Table 6, where "#N&E Feat" refers to the dimensions of node and raw edge features. Table 7 summarizes the skewness of all 13 datasets, where the skewness is measured for the interaction intervals between pairs of nodes in each dataset.

## B.2 Descriptions of Baselines

We select nine baseline link prediction methods, covering a wide range of underlying TGNN architectures: random walk-based TGNN (CAWN), and temporal neighbor-based TGNNs (TGN, EdgeBank, JODIE, DyRep, TGAT, TCL, GraphMixer, DyGFormer).

1. **TGN** maintains an evolving memory for each node in a temporal graph, updating its memory when the node participates in an interaction. The stored historical states of the node are subsequently used by an embedding module to compute its future representation.

Table 6: Statistics of Datasets. The '-' symbol denotes that the dataset does not contain the corresponding feature.

| Datasets | Domains | #Nodes | #Edges | # Unique Edges | #N&E Feat | Duration | Unique Steps | Time Granularity |
|---|---|---|---|---|---|---|---|---|
| CP | Politics | 734 | 74,478 | 51,331 | – & 1 | 14 years | 14 | years |
| CO | Proximity | 694 | 2,426,280 | 79,531 | – & 1 | 1 month | 8,065 | 5 minutes |
| EN | Social | 184 | 125,235 | 3,125 | – & – | 3 years | 22,632 | Unix timestamps |
| FL | Transport | 13,169 | 1,927,145 | 395,072 | – & 1 | 4 months | 122 | days |
| LA | Interaction | 1,980 | 1,293,103 | 154,993 | – & – | 1 month | 1,283,614 | Unix timestamps |
| MO | Interaction | 7,144 | 411,749 | 178,443 | – & 4 | 17 months | 345,600 | Unix timestamps |
| RE | Social | 10,984 | 672,447 | 78,516 | – & 172 | 1 month | 669,065 | Unix timestamps |
| SE | Proximity | 74 | 2,099,519 | 4,486 | – & 2 | 8 months | 565,932 | Unix timestamps |
| UC | Social | 1,899 | 59,835 | 20,296 | – & – | 196 days | 58,911 | Unix timestamps |
| UT | Economics | 255 | 507,497 | 36,182 | – & 1 | 32 years | 32 | years |
| UV | Politics | 201 | 1,035,742 | 31,516 | – & 1 | 72 years | 72 | years |
| US | Politics | 225 | 60,396 | 26,423 | – & 1 | 12 congresses | 12 | congresses |
| WK | Social | 9,227 | 157,474 | 18,257 | – & 172 | 1 month | 152,757 | Unix timestamps |

Table 7: Skewness of datasets measured by interaction intervals.

| Datasets | CP | CO | EN | FL | LA | MO | RE | SE | UC | UT | UV | US | WK |
|---|---|---|---|---|---|---|---|---|---|---|---|---|---|
| Skewness | 1.71 | 10.96 | 4.24 | 2.37 | 3.15 | 4.33 | 2.46 | 18.63 | 5.2 | 6.35 | 6.36 | 9.17 | 4.48 |

2. **EdgeBank** EdgeBank is a transductive link prediction method with no trainable parameters. It stores observed interactions between nodes in a memory unit, which is updated using various strategies. A future interaction is predicted as positive if it is retained in memory, and negative otherwise [21]. Depending on the memory update strategies, EdgeBank has four variants: EdgeBank$_\infty$ uses unlimited memory and retains all observed edges; EdgeBank$_{tw-ts}$ and EdgeBank$_{tw-re}$ retain only recent edges within a fixed-size time window. The window size for EdgeBank$_{tw-ts}$ is set to the duration of the test split, while EdgeBank$_{tw-re}$ adjusts the window size based on the time intervals between repeated edges; EdgeBank$_{th}$ retains only edges that appear more than a specified threshold number of times. We evaluate all four variants and report the best-performing one.

3. **JODIE** is designed for temporal bipartite networks involving user-item interactions. It maintains the states of both user and item nodes and utilizes two coupled recurrent neural networks to update these node states. Additionally, a projection operation is introduced to learn the future representation trajectory for each user and item [13].

4. **DyRep** introduces a recurrent architecture to update node states at each interaction, complemented by a temporal-attentive aggregation module that captures the evolving structural information in temporal graphs. [27].

5. **CAWN** first extracts multiple causal anonymous walks for each node, enabling the exploration of the causality in network dynamics. Then, it employs recurrent neural networks to encode each walk and aggregates them to obtain the final node representation [31].

6. **TGAT** It computes the representation of a node by aggregating its temporal neighbors using the graph attention mechanism [28], with a time encoding function to capture temporal patterns [34].

7. **TCL** first performs a breadth-first search on the temporal subgraph to identify the temporal neighbors of the target node. Subsequently, a graph transformer is used to encode neighbor embeddings and graph topologies, enabling the computation of node representations. Furthermore, a cross-attention mechanism is employed to capture the interdependencies between the two interacting nodes [29].

8. **GraphMixer** utilizes the MLP-Mixer [26] to encode both temporal information and the historical interactions of the target node. Additionally, a node encoder is employed to aggregate the node features of temporal neighbors [7]. In their experiments, they show that a fixed time encoding function outperforms its trainable counterpart.

9. **DyGFormer** is a Transformer-based architecture that computes node representations by aggregating features from each node's temporal neighbors. It introduces a neighbor co-occurrence encoding scheme to capture the correlations between nodes within an interaction and a patching technique to help the model capture long-term dependencies [39].

## B.3 Implementation Details

We train all TGNNs (excluding EdgeBank, which has no trainable parameters) using the Adam optimizer [11], with binary cross-entropy loss as the objective function. TGNNs are trained for 100 epochs with early stopping, where the patience score is set to 20. We use a learning rate of 0.0001 and a batch size of 200 for all methods and datasets. The model with the best validation performance is selected for testing. Each method is run *five* times using random seeds ranging from 0 to 4, with the average performance reported to minimize any deviations. For the hyperparameter $\gamma$ in our LHA module, we search for the best $\gamma$ range from 0.0001 to 1 during the training and validation phases and then use the $\gamma$ with the best validation performance in the test phase. Specifically, we set $\gamma = 0.0001$ for the **MO** and **SE** datasets, $\gamma = 0.1$ for the **UV** dataset, and $\gamma = 0.9$ for the remaining ten datasets. The dimension of historical edge embedding is set equal to the dimension of temporal node embedding, i.e., $d_r = d$, where detailed configurations can be found in Section B.4.

All our experiments are conducted on a GPU server running Ubuntu 22.04, with PyTorch 2.1.0 and CUDA 12.1. We train and test the proposed TAMI framework using a single NVIDIA A100 80G GPU.

## B.4 Configurations of Baselines

[39] conducted an extensive hyperparameter search across all 13 datasets. For consistency, we adopt the optimal hyperparameter settings reported in [39] for all baseline methods. We first outline the configurations that remain consistent across all datasets, followed by the specific hyperparameter settings for each dataset.

The consistent configurations are as follows:

- **JODIE**
  1. Dimension of node memory: 172
  2. Dimension of output representation: 172
  3. Memory updater: vanilla recurrent neural network
- **DyRep**
  1. Dimension of time encoding: 100
  2. Dimension of node memory: 172
  3. Dimension of output representation: 172
  4. Number of graph attention heads: 2
  5. Number of graph convolution layers: 1
  6. Memory updater: vanilla recurrent neural network
- **TGAT**
  1. Dimension of time encoding: 100
  2. Dimension of output representation: 172
  3. Number of graph attention heads: 2
  4. Number of graph convolution layers: 2
- **TGN**
  1. Dimension of time encoding: 100
  2. Dimension of node memory: 172
  3. Dimension of output representation: 172
  4. Number of graph attention heads: 2
  5. Number of graph convolution layers: 1
  6. Memory updater: gated recurrent unit [5]
- **CAWN**
  1. Dimension of time encoding: 100
  2. Dimension of position encoding: 172
  3. Dimension of output representation: 172

4. Number of attention heads for encoding walks: 8

5. Length of each walk (including the target node): 2

6. Time scaling factor $\alpha$: 1e-6

- **TCL**

  1. Dimension of time encoding: 100

  2. Dimension of depth encoding: 172

  3. Dimension of output representation: 172

  4. Number of attention heads: 2

  5. Number of Transformer layers: 2

- **GraphMixer**

  1. Dimension of time encoding: 100

  2. Dimension of output representation: 172

  3. Number of MLP-Mixer layers: 2

  4. Time gap $T$: 2000

- **DyGFormer**

  1. Dimension of time encoding: 100

  2. Dimension of neighbor co-occurrence encoding $d_C$: 50

  3. Dimension of aligned encoding $d$: 50

  4. Dimension of output representation: 172

  5. Number of attention heads: 2

  6. Number of Transformer layers: 2

The hyperparameter settings for each method across different datasets are shown in Table 8, Table 9, and Table 10.

Table 8: Dropout rates of different methods.

| Datasets | JODIE | DyRep | TGAT | TGN | CAWN | TCL | GraphMixer | DyGFormer |
|----------|-------|-------|------|-----|------|-----|------------|-----------|
| CP | 0.0 | 0.0 | 0.2 | 0.3 | 0.0 | 0.2 | 0.2 | 0.1 |
| CO | 0.1 | 0.0 | 0.1 | 0.1 | 0.1 | 0.0 | 0.1 | 0.0 |
| EN | 0.1 | 0.0 | 0.2 | 0.0 | 0.1 | 0.1 | 0.5 | 0.0 |
| FL | 0.1 | 0.1 | 0.1 | 0.1 | 0.1 | 0.1 | 0.2 | 0.1 |
| LA | 0.3 | 0.0 | 0.1 | 0.3 | 0.1 | 0.1 | 0.0 | 0.1 |
| MO | 0.2 | 0.0 | 0.1 | 0.2 | 0.1 | 0.1 | 0.4 | 0.1 |
| RE | 0.1 | 0.1 | 0.1 | 0.1 | 0.1 | 0.1 | 0.5 | 0.2 |
| SE | 0.1 | 0.1 | 0.1 | 0.0 | 0.1 | 0.0 | 0.3 | 0.1 |
| UC | 0.4 | 0.0 | 0.1 | 0.1 | 0.1 | 0.0 | 0.4 | 0.1 |
| UT | 0.4 | 0.1 | 0.1 | 0.2 | 0.1 | 0.0 | 0.1 | 0.0 |
| UV | 0.1 | 0.1 | 0.2 | 0.1 | 0.1 | 0.0 | 0.0 | 0.2 |
| US | 0.2 | 0.0 | 0.1 | 0.1 | 0.1 | 0.3 | 0.4 | 0.0 |
| WK | 0.1 | 0.1 | 0.1 | 0.1 | 0.1 | 0.1 | 0.5 | 0.1 |

## B.5 Skewness of Temporal Difference

Table 11 presents the skewness of temporal difference for both the baseline (Original TE) and the proposed (LTE) time encoding functions. The skewness score is computed using Fisher's moment coefficient of skewness [12, 2]. A positive skewness score indicates that the temporal differences are right-skewed, a negative score indicates they are left-skewed, and a score of 0 suggests a balanced distribution. Compared to the baseline encoding function, the proposed LTE method effectively reduces the skewness of temporal difference.

## C  Comprehensive Results for Transductive Temporal Link Prediction

This section presents additional results for transductive link prediction. Section C.1 presents the comprehensive results of Table 1. Section C.2 shows that applying LTE and LHA individually also improves the link prediction accuracy of the integrated model. Section C.3 discusses how the

Table 9: The sample size of temporal neighbors, the number of causal anonymous walks, and the length of input sequences & the patch size of different methods. The number in parentheses indicates the number of historical interactions used to compute each temporal node embedding during the test stage. For example, on the CP dataset, setting the input sequence length to 2048 denotes that, on average, 99.98% of historical interactions are used to compute a single temporal node embedding.

| Datasets | JODIE | TGAT | TGN | CAWN | TCL | GraphMixer | DyGFormer |
|---|---|---|---|---|---|---|---|
| CP | 10 | 20 | 10 | 128 | 20 | 20 | 2048 (99.98%) & 64 |
| CO | 10 | 20 | 10 | 64 | 20 | 20 | 32 (0.49%) & 1 |
| EN | 10 | 20 | 10 | 32 | 20 | 20 | 256 (31.69%) & 8 |
| FL | 10 | 20 | 10 | 64 | 20 | 20 | 256 (34.93%) & 8 |
| LA | 10 | 20 | 10 | 128 | 20 | 10 | 512 (41.23%) & 16 |
| MO | 10 | 20 | 10 | 64 | 20 | 20 | 256 (57.59%) & 8 |
| RE | 10 | 20 | 10 | 32 | 20 | 10 | 64 (40.48%)& 2 |
| SE | 10 | 20 | 10 | 64 | 20 | 20 | 32 (0.05%) & 1 |
| UC | 10 | 20 | 10 | 64 | 20 | 20 | 32 (34.13%) & 1 |
| UT | 10 | 20 | 10 | 64 | 20 | 20 | 256 (7.05%) & 8 |
| UV | 10 | 20 | 10 | 64 | 20 | 20 | 128 (1.66%) & 4 |
| US | 10 | 20 | 10 | 32 | 20 | 20 | 256 (63.03%) & 8 |
| WK | 10 | 20 | 10 | 32 | 20 | 30 | 32 (39.37%) & 1 |

Table 10: Strategies for sampling temporal neighbors during testing and the best-performing variants of EdgeBank.

| Datasets | DyRep | TGAT | TGN | TCL | GraphMixer | EdgeBank Variant |
|---|---|---|---|---|---|---|
| CP | uniform | uniform | uniform | uniform | uniform | EdgeBank$_{tw-ts}$ |
| CO | recent | recent | recent | recent | recent | EdgeBank$_{tw-re}$ |
| EN | recent | recent | recent | recent | recent | EdgeBank$_{tw-ts}$ |
| FL | recent | recent | recent | recent | recent | EdgeBank$_\infty$ |
| LA | recent | recent | recent | recent | recent | EdgeBank$_{tw-ts}$ |
| MO | recent | recent | recent | recent | recent | EdgeBank$_{tw-ts}$ |
| RE | recent | uniform | recent | uniform | recent | EdgeBank$_\infty$ |
| SE | recent | recent | recent | recent | recent | EdgeBank$_{th}$ |
| UC | recent | recent | recent | recent | recent | EdgeBank$_\infty$ |
| UT | recent | uniform | recent | uniform | uniform | EdgeBank$_{tw-re}$ |
| UV | recent | recent | uniform | uniform | uniform | EdgeBank$_{tw-re}$ |
| US | recent | recent | recent | uniform | recent | EdgeBank$_{tw-ts}$ |
| WK | recent | recent | recent | recent | recent | EdgeBank$_\infty$ |

Table 11: Skewness of temporal difference in different time encoding functions on 13 datasets. The proposed LTE time encoding function effectively reduces the skewness of temporal difference.

| Datasets | CP | CO | EN | FL | LA | MO | RE | SE | UC | UT | UV | US | WK |
|---|---|---|---|---|---|---|---|---|---|---|---|---|---|
| Original TE | 1.73 | 28.81 | 6.35 | 7.45 | 32.76 | 4.14 | 3.47 | 140.42 | 2.38 | 0.34 | 0.62 | 9.22 | 2.64 |
| LTE | 0.69 | 1.96 | -0.60 | 2.37 | -0.61 | 0.30 | -0.31 | 0.07 | -1.14 | -0.96 | -0.95 | 9.22 | -0.8865 |

effectiveness of LTE is influenced by the skewness of temporal differences in a dataset and the configurations of underlying TGNNs. Section C.4 explores the impact of the hyperparameter $\gamma$ in our LHA module. Section C.5 studies the robustness of TAMI under different aggregation strategies and values of $k$ in our LHA module. Section C.6 highlights that TGNNs suffer from the loss of interaction histories, which reduces their link prediction performance. Our LHA module helps mitigate this loss and improves TGNNs' performance. Section C.7 evaluates the effectiveness of LHA under conditions of limited memory.

## C.1 Comprehensive Results of Table 1

Table 12 presents the comprehensive results of Table 1. Results show that TAMI consistently improves the link prediction accuracy of underlying TGNNs and remains effective under different negative sampling strategies.

## C.2 Adaptivity of LTE and LHA to Various Types of TGNNs

Table 13 demonstrates the comprehensive results of Table 5 in Section 4.5. It also presents the link prediction performance of TGNNs integrated with the proposed LTE and LHA, alongside their vanilla

Table 12: AP for transductive link prediction. Negative edges are generated using the random (rnd), historical (hist), and inductive (ind) negative sampling strategies proposed in [21]. NSS stands for negative sampling strategies. Standard deviations are summarized in Table 21.

| NSS | Methods | CP | CO | EN | FL | LA | MO | RE | SE | UC | UT | UV | US | WK |
|---|---|---|---|---|---|---|---|---|---|---|---|---|---|---|
| rnd | JODIE | 69.26 | 95.31 | 84.77 | 95.60 | 70.85 | 80.23 | 98.31 | 89.89 | 89.43 | 64.94 | 63.91 | 75.05 | 96.50 |
| | DyRep | 66.54 | 95.98 | 82.38 | 95.29 | 71.92 | 81.97 | 98.22 | 88.87 | 65.14 | 63.21 | 62.81 | 75.34 | 94.86 |
| | TGAT | 70.73 | 96.28 | 71.12 | 94.03 | 73.42 | 85.84 | 98.52 | 93.16 | 79.63 | 61.47 | 52.21 | 68.52 | 96.94 |
| | TGN | 70.88 | 96.89 | 86.53 | 97.95 | 77.07 | **89.15** | 98.63 | 93.57 | 92.34 | 65.03 | **65.72** | **75.99** | 98.45 |
| | CAWN | 69.82 | 90.26 | 89.56 | 98.51 | 86.99 | 80.15 | 99.11 | 84.96 | 95.18 | 65.39 | 52.84 | 70.58 | 98.76 |
| | EdgeBank | 64.55 | 92.58 | 83.53 | 89.35 | 79.29 | 57.97 | 94.86 | 74.95 | 76.20 | 60.41 | 58.49 | 58.39 | 90.37 |
| | TCL | 68.67 | 92.44 | 79.70 | 91.23 | 67.27 | 82.38 | 97.53 | 93.13 | 89.57 | 62.21 | 51.90 | 69.59 | 96.47 |
| | GraphMixer | 75.90 | 91.94 | 82.26 | 90.98 | 75.56 | 82.83 | 97.33 | 93.34 | 93.38 | 62.61 | 52.20 | 71.55 | 97.23 |
| | DyGFormer | 97.91 | 98.31 | 92.46 | 98.92 | 93.01 | 87.66 | 99.22 | 94.66 | 95.66 | 65.07 | 55.88 | 70.44 | 99.02 |
| | **with TAMI** | | | | | | | | | | | | | |
| | GraphMixer | 78.38 | 95.26 | 90.97 | 96.75 | 88.13 | 83.53 | 98.84 | 93.41 | 96.20 | 62.98 | 57.74 | 71.57 | 98.89 |
| | Imp. (%) | 3.27% | 3.61% | 10.59% | 6.34% | 16.64% | 0.85% | 1.56% | 0.07% | 3.02% | 0.59% | 10.61% | 0.03% | 1.71% |
| | DyGFormer | 98.67 | 98.70 | 92.66 | 98.94 | 94.03 | 88.49 | 99.29 | 94.74 | 96.72 | 66.39 | 56.02 | 71.40 | 99.25 |
| | Imp. (%) | 0.78% | 0.40% | 0.22% | 0.02% | 1.10% | 0.95% | 0.07% | 0.08% | 1.11% | 2.03% | 0.25% | 1.36% | 0.23% |
| hist | JODIE | 51.79 | 95.31 | 69.85 | 66.48 | 74.35 | 78.94 | 80.03 | 87.44 | 75.24 | 61.39 | 70.02 | 51.71 | 83.01 |
| | DyRep | 63.31 | 96.39 | 71.19 | 67.61 | 74.92 | 75.60 | 79.83 | 93.29 | 55.10 | 59.19 | 69.30 | **86.88** | 79.93 |
| | TGAT | 67.13 | 96.05 | 64.07 | 72.38 | 71.59 | 82.19 | 79.55 | 95.01 | 68.27 | 55.74 | 52.96 | 62.14 | 87.38 |
| | TGN | 68.42 | 93.05 | 73.91 | 66.70 | 76.87 | **87.06** | 81.22 | 94.45 | 80.43 | 58.44 | 69.37 | 74.00 | 86.86 |
| | CAWN | 66.53 | 84.16 | 64.73 | 64.72 | 69.86 | 74.05 | 80.82 | 85.53 | 65.30 | 55.71 | 51.26 | 68.82 | 71.21 |
| | EdgeBank | 63.84 | 88.81 | 76.53 | 70.53 | 73.03 | 60.71 | 73.59 | 80.57 | 65.50 | **81.32** | **84.89** | 63.20 | 73.35 |
| | TCL | 65.93 | 93.86 | 70.66 | 70.68 | 59.30 | 77.06 | 77.14 | 94.74 | 76.20 | 55.90 | 52.30 | 80.53 | 89.05 |
| | GraphMixer | 74.34 | 93.29 | 77.98 | 71.47 | 72.47 | 77.77 | 78.44 | 94.93 | 84.11 | 57.05 | 51.20 | 81.65 | 90.90 |
| | DyGFormer | 97.00 | 97.57 | 75.63 | 66.59 | 81.57 | 85.85 | 81.57 | 97.38 | 82.17 | 64.41 | 60.84 | 85.30 | 82.23 |
| | **with TAMI** | | | | | | | | | | | | | |
| | GraphMixer | 78.81 | 93.30 | **81.68** | **73.01** | 80.23 | 83.61 | 82.56 | 96.80 | **87.69** | 69.74 | 70.90 | 84.56 | **90.97** |
| | Imp. (%) | 6.01% | 0.01% | 4.74% | 2.15% | 10.71% | 7.51% | 5.25% | 1.97% | 4.26% | 22.24% | 38.48% | 3.56% | 0.08% |
| | DyGFormer | 98.96 | 97.72 | 81.02 | 67.77 | 83.40 | 86.26 | 85.18 | 97.56 | 85.89 | 65.16 | 81.72 | 86.10 | 82.38 |
| | Imp. (%) | 2.02% | 0.15% | 7.13% | 1.77% | 2.24% | 0.48% | 4.43% | 0.18% | 4.53% | 1.16% | 34.32% | 0.94% | 0.18% |
| ind | JODIE | 48.42 | 93.43 | 68.96 | 69.07 | 62.67 | 65.23 | 86.96 | 89.82 | 65.99 | 60.42 | 67.79 | 50.27 | 75.65 |
| | DyRep | 58.61 | 94.18 | 67.79 | 70.57 | 64.41 | 61.66 | 86.30 | 93.28 | 54.79 | 60.19 | 67.53 | **83.44** | 70.21 |
| | TGAT | 68.82 | 94.35 | 63.94 | 75.48 | 71.13 | 75.95 | 89.59 | 94.84 | 68.67 | 60.61 | 52.89 | 61.91 | 87.00 |
| | TGN | 65.34 | 90.18 | 70.89 | 71.09 | 65.95 | 77.50 | 88.10 | 95.13 | 70.94 | 61.04 | 67.63 | 67.57 | 85.62 |
| | CAWN | 67.75 | 89.31 | 75.15 | 69.18 | 67.48 | 73.51 | 91.67 | 88.32 | 64.61 | 62.54 | 52.19 | 65.81 | 74.06 |
| | EdgeBank | 62.16 | 85.20 | 73.89 | 81.08 | 75.49 | 49.43 | 85.48 | 83.69 | 57.43 | 72.97 | 66.30 | 64.74 | 80.63 |
| | TCL | 65.85 | 91.35 | 71.29 | 74.62 | 58.21 | 74.65 | 87.45 | 94.90 | 76.01 | 61.06 | 50.62 | 78.15 | 86.76 |
| | GraphMixer | 69.48 | 90.87 | 75.01 | 74.87 | 68.12 | 74.26 | 85.26 | 94.72 | 80.10 | 60.15 | 51.60 | 79.63 | 88.59 |
| | DyGFormer | 95.44 | 94.75 | 77.41 | 70.92 | 73.97 | 81.24 | 91.11 | 97.68 | 72.25 | 55.79 | 51.91 | 81.25 | 78.29 |
| | **with TAMI** | | | | | | | | | | | | | |
| | GraphMixer | 70.94 | 96.12 | **88.95** | **93.64** | **91.06** | 79.82 | **96.19** | 96.09 | **84.12** | **87.73** | **79.53** | 83.31 | **93.89** |
| | Imp. (%) | 2.10% | 5.78% | 18.58% | 25.07% | 33.68% | 7.49% | 12.82% | 1.45% | 5.02% | 45.85% | 54.13% | 4.62% | 5.98% |
| | DyGFormer | 97.25 | 98.47 | 86.23 | 75.55 | 74.03 | 92.39 | 94.37 | 97.76 | 80.13 | 68.01 | 78.19 | 81.31 | 78.96 |
| | Imp. (%) | 1.90% | 3.93% | 11.39% | 6.53% | 0.08% | 13.72% | 3.58% | 0.08% | 10.91% | 21.90% | 50.63% | 0.07% | 0.86% |

counterparts. The results suggest that TAMI is adaptable to different types of TGNNs, consistently improving the performance of underlying models. Moreover, applying LTE and LHA individually also improves the link prediction accuracy of the integrated model.

## C.3 Application Scenarios of LTE

The effectiveness of the proposed LTE is influenced by the distribution of temporal distances within the dataset. Based on the skewness of the temporal differences in the original TE, we classify the datasets into two types: (1) skewed, where small temporal differences dominate, while a considerable number of large temporal differences exist; and (2) balanced, where the temporal differences roughly follow a normal distribution. As shown in Table 3 and Table 11, for balanced datasets (e.g., UV and UT), the improvements observed in GraphMixer are marginal. This is because LTE is specifically designed to address the skewness of temporal differences, which is rarely presented in balanced datasets. For datasets with skewed temporal differences (e.g., CP, EN, and UC), LTE significantly improves GraphMixer's performance compared to the vanilla version with the original TE.

Table 13: AP for transductive setting. The proposed LTE and LHA can be integrated into various types of TGNNs. When applied individually, both LTE and LHA improve the link prediction accuracy of the integrated models. No LTE and TAMI results are reported for JODIE because it does not utilize the time encoding function.

| Method | EN | LA | UC | UV |
|---|---|---|---|---|
| TGN | $87.09 \pm 1.04$ | $75.73 \pm 1.53$ | $91.87 \pm 1.42$ | $65.68 \pm 1.19$ |
| TGAT | $72.89 \pm 1.12$ | $73.37 \pm 0.07$ | $79.29 \pm 0.06$ | $53.14 \pm 0.56$ |
| CAWN | $88.39 \pm 0.07$ | $86.99 \pm 0.01$ | $94.97 \pm 0.08$ | $52.88 \pm 0.08$ |
| JODIE | $84.79 \pm 4.80$ | $70.17 \pm 3.82$ | $89.15 \pm 0.98$ | $63.52 \pm 0.10$ |
| **w/ LTE** | | | | |
| TGN | $90.23 \pm 0.47$ | $85.03 \pm 1.45$ | $94.17 \pm 1.12$ | $64.90 \pm 0.32$ |
| Imp. (%) | 3.61% | 12.27% | 2.51% | -1.20% |
| TGAT | $84.75 \pm 0.04$ | $83.49 \pm 0.01$ | $94.85 \pm 0.50$ | $54.97 \pm 3.96$ |
| Imp. (%) | 16.27% | 13.79% | 19.62% | 3.45% |
| CAWN | $90.70 \pm 0.39$ | $88.76 \pm 0.03$ | $96.63 \pm 0.08$ | $52.92 \pm 0.77$ |
| Imp. (%) | 2.61% | 2.03% | 1.75% | 0.08% |
| **w/ LHA** | | | | |
| TGN | $90.20 \pm 0.60$ | $90.12 \pm 0.30$ | $93.34 \pm 0.55$ | $67.95 \pm 1.22$ |
| Imp. (%) | 3.57% | 19.00% | 1.61% | 3.46% |
| TGAT | $88.06 \pm 0.12$ | $87.97 \pm 0.23$ | $89.74 \pm 0.11$ | $57.58 \pm 0.27$ |
| Imp. (%) | 20.81% | 19.89% | 13.17% | 8.37% |
| CAWN | $90.13 \pm 0.01$ | $89.29 \pm 0.04$ | $95.16 \pm 0.03$ | $56.50 \pm 0.54$ |
| Imp. (%) | 1.96% | 2.64% | 0.21% | 6.85% |
| JODIE | $90.62 \pm 0.01$ | $87.95 \pm 0.01$ | $92.44 \pm 0.02$ | $65.57 \pm 0.34$ |
| Imp. (%) | 6.88% | 25.33% | 3.68% | 3.22% |
| **w/ TAMI** | | | | |
| TGN | $92.34 \pm 0.04$ | $92.83 \pm 0.01$ | $95.36 \pm 0.11$ | $67.80 \pm 0.59$ |
| Imp. (%) | 6.03% | 22.58% | 3.80% | 3.22% |
| TGAT | $91.37 \pm 0.04$ | $91.60 \pm 0.13$ | $96.36 \pm 0.18$ | $60.03 \pm 0.34$ |
| Imp. (%) | 25.35% | 24.85% | 21.53% | 12.98% |
| CAWN | $91.23 \pm 0.06$ | $91.02 \pm 0.00$ | $96.69 \pm 0.13$ | $57.49 \pm 0.18$ |
| Imp. (%) | 3.21% | 4.64% | 1.81% | 8.72% |

## C.4   Analysis of the Hyperparameter $\gamma$ in LHA

In this experiment, we study the impact of the hyperparameter $\gamma$ in our LHA module on link prediction performance. We train, validate, and test models with $\gamma = (0.1, 0.3, 0.5, 0.7, 0.9, 1)$ and report the test AP in Table 14. As shown, integrating GraphMixer into TAMI consistently improves its performance over its vanilla counterpart across all values of $\gamma$. In addition, $\gamma$ affects link prediction accuracy differently across different datasets. For the UV dataset, the performance of GraphMixer with LHA remains similar across different values of $\gamma$. When it comes to the EN, LA, and UC datasets, higher $\gamma$ values generally yield better performance. This is because larger $\gamma$ values place more weight on recent historical interactions, while reducing the influence of older ones. These results suggest that prioritizing recent interactions is more advantageous for these datasets.

Table 14: AP of GraphMixer with varying values of the hyperparameter $\gamma$ when integrated with our LHA module. The first and the second best performances are marked in **bold** and underlined respectively.

| | EN | LA | UC | UV |
|---|---|---|---|---|
| GraphMixer | 82.26 | 75.56 | 93.38 | 52.20 |
| **with TAMI** | | | | |
| $\gamma = 0.1$ | 90.41 | 85.97 | 95.80 | **58.02** |
| $\gamma = 0.3$ | 90.87 | 87.42 | 96.11 | 57.99 |
| $\gamma = 0.5$ | **90.97** | 87.94 | 96.22 | 57.95 |
| $\gamma = 0.7$ | 90.96 | 88.18 | **96.24** | 57.87 |
| $\gamma = 0.9$ | 90.77 | **88.19** | 96.20 | 57.74 |
| $\gamma = 1$ | 90.60 | 88.09 | 96.16 | 57.55 |

## C.5 Analysis of the Choice of Aggregation Strategies and the Value of $k$ in LHA

To predict the future link between two nodes, LHA retrieves all stored $k$ historical edge embeddings for the target node pair and summarizes them into a single vector using an aggregation function. In this experiment, we study the effectiveness of TAMI under different values of $k$ and aggregation strategies. By default, we set $k = 1$ and use the **most-recent** aggregator. Note that our LHA module uses an exponentially weighted moving average to compute the dedicated historical edge embeddings for a given target pair of nodes, as shown in Equation 4. Thus, even if $k = 1$, LHA considers all the interaction history between the pair of nodes.

In Table 15, we show the performance of TAMI with different $k$ values and aggregation functions. The **mean** aggregator computes the average of the $k$ historical edge embeddings, whereas the **max** aggregator follows the implementation in GraphSAGE [8], where each historical edge embedding is passed through a fully connected neural network, followed by an element-wise max-pooling operation across all transformed vectors. Overall, the performance of different variants of TAMI is better than the vanilla version (without TAMI), indicating the effectiveness of our design. It is also important to note that larger $k$ value does not necessarily lead to better performance. This is expected as different datasets may have different levels of dependency on the interaction history for temporal link prediction. In addition, TAMI is not sensitive to the choice of aggregation functions.

Table 15: Test AP of DyGFormer on the EN and CP datasets with different $k$ values and aggregation strategies.

| Method | EN | CP |
|---|---|---|
| DyGFormer (Vanilla) | 92.46 | 97.91 |
| with TAMI most-recent | 92.66 | 98.67 |
| mean ($k = 1$) | 92.66 | 98.67 |
| mean ($k = 2$) | 92.60 | 98.76 |
| mean ($k = 3$) | 92.56 | **98.77** |
| max ($k = 1$) | 92.66 | 98.67 |
| max ($k = 2$) | **92.84** | 98.74 |
| max ($k = 3$) | 92.60 | 98.74 |

## C.6 Preventing the Forgetting of Interaction Histories by LHA

In Figure 5, we demonstrate that the loss of interaction histories occurs in GraphMixer [7] and degrades its link prediction performance. Here we extend our analysis to JODIE [13]. JODIE adopts two RNNs to maintain an evolving temporal memory for each node and uses stored historical node states to compute temporal node embeddings. When predicting the link between two nodes, if neither node appears in the other node's 20 most recent interactions, we consider their interaction histories as no longer retained in their temporal node embeddings. Figure 8 presents the distribution of the appearance index along with the link prediction accuracy. The appearance index represents the earliest position at which at least one node appears in the interaction sequence of the other. For instance, an index value of 1 means that at least one node is the most recent encounter of the other, while a larger index indicates that fewer historical interactions between the nodes are retained. The term "$> 20$" denotes the case where neither node appears in the most recent 20 interactions of the other and their interaction histories are forgotten.

As shown in Figure 8, the "$> 20$" term has the highest percentage across all examined datasets. This suggests that the loss of historical interactions also occurs in memory-based TGNNs. On the EN, LA, and UC datasets, the AP of the vanilla JODIE model decreases as the appearance index increases, indicating that retaining fewer historical interactions leads to worse link prediction performance. In comparison, when the proposed LHA is incorporated, the AP of JODIE consistently improves across all datasets, demonstrating the effectiveness of our LHA module in mitigating the loss of historical interactions.

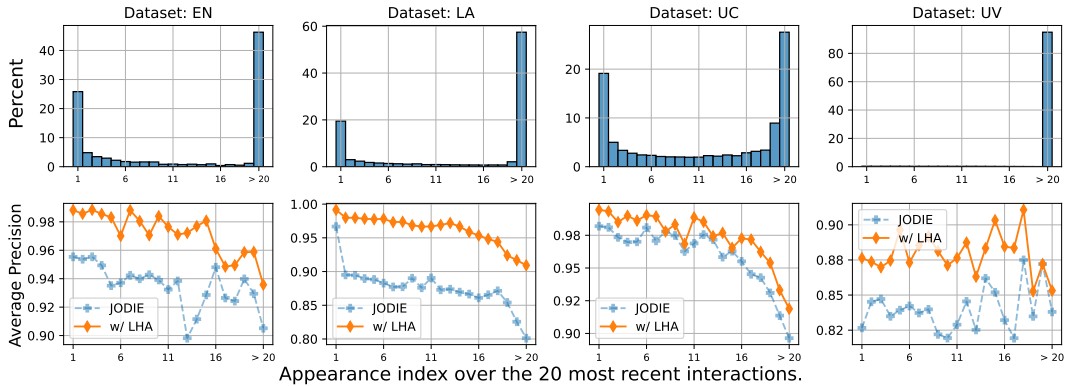

Figure 8: AP of JODIE to the number of interaction histories retained. An increase in the appearance index indicates that fewer interaction histories are retained for predicting future links.

## C.7 Cold Start of LHA in Inference

LHA maintains the historical connections between nodes. It may happen that in the training set, two nodes never interact. In this experiment, we evaluate how LHA performs in that case. Specifically, we introduce a variant called "w/ LHA (no history before test)", where the interaction histories accumulated during the training and validation phases are not loaded when the test stage begins. In other words, interaction histories must be constructed from scratch, as the test progresses. This variant is compared against the default LHA setting, "w/ LHA," where the history accumulated during training and validation is loaded before testing. Figure 9 illustrates the test AP of various methods. The x-axis shows the test phase timeline, with timestamps normalized to [0, 1] for better visualization.

First, integrating LHA into GraphMixer and JODIE consistently boosts their performance throughout the testing phase, surpassing their vanilla counterparts. Second, even when LHA history is not initialized at the beginning of the test phase (i.e., in the "w/ LHA (no history before test)" variant), the augmented TGNNs still outperform their vanilla versions on the EN, LA, and UC datasets. This indicates that the LHA module provides significant performance improvements, even with a limited history in the early stage of inference. As the test progresses and more interaction histories are stored in the LHA, the performance of the "w/ LHA (no history before test)" variant gradually converges to that of the "w/ LHA" variant. This is expected, as both variants share increasingly similar histories, leading to similar performance over time.

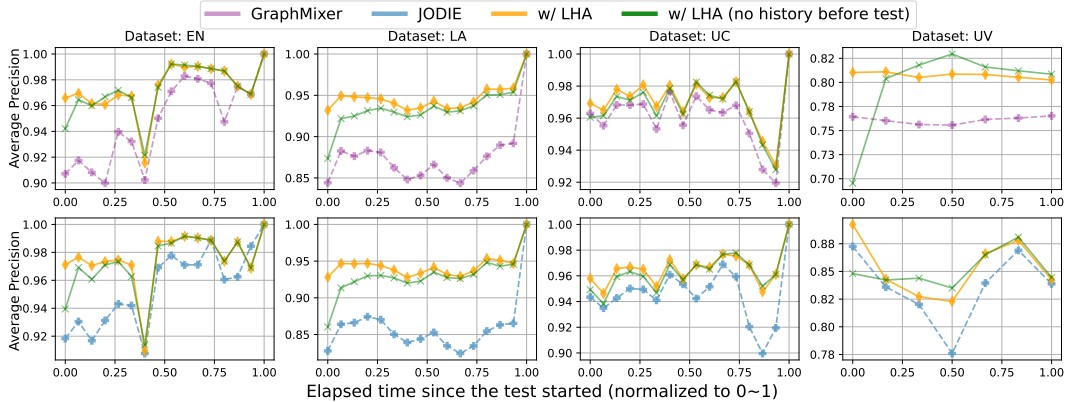

Figure 9: LHA consistently improves the performance of TGNNs throughout the entire testing phase. The "w/ LHA" variant denotes that interaction histories accumulated during the training and validation phases are loaded at the beginning of the test phase. In contrast, the "w/ LHA (no history before test)" variant starts the test phase without any interaction histories stored in the LHA memory module. Test node pairs are grouped into chronological bins, and the average link prediction accuracy for each bin is reported.

Table 16: AP for inductive link prediction. Imp. (%) denotes the percentage of improvement. Improved results are colored in blue. The first and the second best performances are marked in **bold** and underlined respectively. For dataset names, please kindly refer to Section B.1.

| Methods | CP | CO | EN | FL | LA | MO | RE |
|---|---|---|---|---|---|---|---|
| JODIE | 53.92 ± 0.94 | 94.34 ± 1.45 | 80.67 ± 2.11 | 94.74 ± 0.37 | 78.31 ± 3.82 | 79.63 ± 1.92 | 96.50 ± 0.13 |
| DyRep | 54.02 ± 0.76 | 92.18 ± 0.41 | 74.55 ± 3.95 | 92.88 ± 0.73 | 83.02 ± 1.48 | 81.07 ± 0.44 | 96.09 ± 0.11 |
| TGAT | 55.18 ± 0.79 | 95.87 ± 0.11 | 67.63 ± 0.96 | 88.73 ± 0.33 | 78.48 ± 0.07 | 85.50 ± 0.19 | 97.09 ± 0.04 |
| TGN | 54.10 ± 0.93 | 93.82 ± 0.99 | 78.74 ± 1.53 | 95.03 ± 0.60 | 80.05 ± 4.00 | **89.04** ± 1.17 | 97.50 ± 0.07 |
| CAWN | 55.80 ± 0.69 | 89.55 ± 0.30 | 84.97 ± 0.21 | 97.60 ± 0.02 | 89.48 ± 0.02 | 81.42 ± 0.24 | 98.62 ± 0.01 |
| TCL | 54.30 ± 0.66 | 91.11 ± 0.12 | 76.14 ± 0.79 | 83.41 ± 0.07 | 73.53 ± 1.66 | 80.60 ± 0.22 | 94.09 ± 0.07 |
| GraphMixer | 57.46 ± 0.11 | 90.61 ± 0.09 | 75.94 ± 0.21 | 83.00 ± 0.04 | 82.06 ± 0.25 | 81.35 ± 0.01 | 95.23 ± 0.03 |
| DyGFormer | 88.14 ± 0.19 | 98.06 ± 0.01 | 89.86 ± 0.28 | 97.79 ± 0.01 | 94.20 ± 0.17 | 87.24 ± 0.35 | 98.83 ± 0.03 |
| with TAMI | | | | | | | |
| GraphMixer | 58.57 ± 0.92 | 94.49 ± 0.21 | 85.54 ± 0.45 | 92.64 ± 0.06 | 90.71 ± 0.64 | 81.55 ± 0.10 | 97.93 ± 0.01 |
| Imp. (%) | 1.93% | 4.28% | 12.63% | 11.62% | 10.55% | 0.25% | 2.84% |
| DyGFormer | **88.97** ± 0.41 | **98.30** ± 0.00 | **89.87** ± 0.06 | **97.85** ± 0.02 | **95.08** ± 0.13 | 88.08 ± 0.23 | **98.91** ± 0.00 |
| Imp. (%) | 0.94% | 0.24% | 0.01% | 0.06% | 0.93% | 0.96% | 0.08% |

| Methods | SE | UC | UT | UV | US | WK |
|---|---|---|---|---|---|---|
| JODIE | 91.96 ± 0.48 | 79.27 ± 1.97 | 59.65 ± 0.77 | **56.86** ± 0.06 | 54.93 ± 2.29 | 94.82 ± 0.20 |
| DyRep | 90.04 ± 0.47 | 57.48 ± 1.87 | 57.02 ± 0.69 | 53.62 ± 2.22 | 57.28 ± 0.71 | 92.43 ± 0.37 |
| TGAT | 91.41 ± 0.16 | 79.11 ± 0.28 | 61.03 ± 0.18 | 51.98 ± 0.14 | 51.00 ± 3.11 | 96.22 ± 0.07 |
| TGN | 90.77 ± 0.86 | 86.66 ± 1.83 | 58.31 ± 3.15 | 54.13 ± 2.37 | **58.63** ± 0.37 | 97.83 ± 0.04 |
| CAWN | 79.94 ± 0.18 | 92.45 ± 0.06 | **65.24** ± 0.21 | 49.29 ± 0.87 | 53.17 ± 1.20 | 98.24 ± 0.03 |
| TCL | 91.55 ± 0.09 | 87.36 ± 2.03 | 62.21 ± 0.12 | 51.60 ± 0.97 | 52.59 ± 0.97 | 96.22 ± 0.17 |
| GraphMixer | 91.71 ± 0.13 | 91.39 ± 0.05 | 62.26 ± 0.13 | 50.58 ± 0.45 | 50.72 ± 0.64 | 96.47 ± 0.13 |
| DyGFormer | 93.16 ± 0.08 | 94.40 ± 0.21 | 63.86 ± 0.35 | 56.16 ± 0.07 | 54.93 ± 0.53 | 98.54 ± 0.07 |
| with TAMI | | | | | | |
| GraphMixer | 93.03 ± 0.46 | 93.44 ± 0.29 | 62.40 ± 1.50 | 56.63 ± 0.36 | 51.44 ± 0.75 | 98.17 ± 0.01 |
| Imp. (%) | 1.44% | 2.24% | 0.22% | 11.96% | 1.42% | 1.76% |
| DyGFormer | **93.17** ± 0.01 | **95.67** ± 0.09 | 64.25 ± 0.24 | 56.23 ± 0.17 | 55.08 ± 0.21 | **98.88** ± 0.07 |
| Imp. (%) | 0.01% | 1.35% | 0.61% | 0.12% | 0.27% | 0.35% |

# D  Comprehensive Results for Inductive Temporal Link Prediction

This section presents comprehensive results for inductive link prediction. Section D.1 demonstrates that LTE and LHA improve the link prediction performance of integrated models in the inductive setting. Section D.2 presents the results of an ablation study in the inductive setting, showing that both LTE and LHA remain effective. In Section D.3, we show that LTE and LHA are robust to an increasing number of negative links per positive link in the inductive setting. Finally, Section D.4 shows that the proposed LTE and LHA can be integrated into various types of TGNN while enhancing their performance in the inductive setting.

## D.1  Main Results for Inductive Link Prediction

Table 16 summarizes the test performance of methods in the inductive setting. The results demonstrate that TAMI improves GraphMixer and DyGFormer on all the 13 datasets. This highlights the effectiveness and versatility of TAMI, suggesting TAMI can enhance link prediction performance in inductive settings.

## D.2  Ablation Study for Inductive Link Prediction

In this experiment, we study the individual contributions of LTE and LHA to the performance improvements of the integrated TGNNs. Table 17 presents the test performance of TAMI and its two variants: **w/ LTE**, where we replace the original TE in TGNNs with the proposed LTE and keep the rest unchanged; **w/ LHA**, where we integrate the LHA module into TGNNs and keep the rest unchanged.

First, replacing the original time encoding function with the proposed LTE improves the performance of GraphMixer on 11 datasets and the performance of DyGFormer on 12 datasets. This suggests that LTE can enhance the link prediction performance of integrated models in the inductive setting.

Second, incorporating the LHA module improves the performance of TGNNs. In the inductive setting, even though no test edges are observed during training (i.e., no historical interactions are stored in the LHA memory at the beginning of the test phase), the LHA module still improves the performance of GraphMixer and DyGFormer on 12 datasets. This is because LHA can update its memory as the test stage progresses and improve the accuracy of subsequent link predictions. These results suggest that the LHA unit remains effective in the inductive setting.

### D.3 Robustness to the Increase of Negative Links (Inductive)

In Section 4.4, we demonstrate that TAMI is robust to an increasing ratio of negative links to positive links under the transductive setting. In this experiment, we evaluate the robustness of TAMI in the inductive setting. Table 18 presents the test performance of TGNNs under the inductive setting. The term "NEG=50" denotes that the connection probability of each positive link is evaluated against 50 negative links. The results demonstrate that the performance of underlying TGNNs consistently improves across various numbers of negative links per positive link on all examined datasets. These findings suggest that the proposed TAMI framework remains effective against the increasing ratio of negative links in the inductive setting.

### D.4 Adaptivity to Different TGNN Architectures (Inductive)

In Section 4.5 and Section C.2, we demonstrate that the proposed TAMI, LTE, and LHA can enhance the link prediction performance of underlying TGNNs in the transductive setting. In this experiment, we explore whether the proposed methods can also improve the performance of TGNNs in the inductive setting. Table 19 presents the performance of the vanilla TGNNs along with those enhanced with the proposed modules. No results for w/ LTE and w/ TAMI are reported for JODIE, as it does not incorporate the time encoding function. The results demonstrate that LTE, LHA, and TAMI consistently improve the inductive AP of TGNNs. This suggests TAMI and the two novel modules LTE and LHA remain effective in the inductive setting and can improve the performance of various TGNNs.

## E  Experimental Settings on the TGB Datasets

We further evaluate TAMI on three large-scale datasets from the Temporal Graph Benchmark (TGB) [9], including tgbl-wiki-v2, tgbl-review-v2, and tgbl-coin-v2. Dataset statistics are available at https://tgb.complexdatalab.com/docs/linkprop/. We apply TAMI to the recent state-of-the-art method DyGFormer and compare its performance against the original version. For fair comparison, we adopt the best hyperparameter settings of DyGFormer on the TGB benchmark, as detailed in [36]. In our LHA module, the hyperparameter $\gamma$ is fixed at 0.9 across the training, validation, and testing phases. The dimension of historical edge embeddings is equal to the dimension of temporal node embeddings. The detailed results are presented in Table 2. We also report the original skewness of temporal differences in the TGB datasets and the skewness after applying our LTE in Table 20 below. Results show that our LTE effectively reduces the skewness in the distribution of the temporal differences.

## F  Standard Deviations in Main Text

Table 21, Table 22, and Table 23 present the standard deviations of the results shown in Table 1, Table 3, and Table 4 in the main text, respectively.

Table 17: Ablation study. AP for inductive link prediction. For dataset names, please kindly refer to Section B.1.

| Methods | CP | CO | EN | FL | LA | MO | RE |
|---|---|---|---|---|---|---|---|
| GraphMixer | 57.46 ± 0.11 | 90.61 ± 0.09 | 75.94 ± 0.21 | 83.00 ± 0.04 | 82.06 ± 0.25 | 81.35 ± 0.01 | 95.23 ± 0.03 |
| w/ LTE | 58.25 ± 1.14 | 90.83 ± 0.27 | 76.15 ± 0.26 | 83.05 ± 0.18 | 81.22 ± 0.90 | 81.73 ± 0.18 | 95.31 ± 0.01 |
| Imp. (%) | 1.37% | 0.25% | 0.27% | 0.07% | -1.02% | 0.47% | 0.08% |
| w/ LHA | 57.30 ± 0.30 | 95.03 ± 1.06 | 85.16 ± 0.15 | 92.60 ± 0.06 | 90.92 ± 0.24 | 81.50 ± 0.06 | 97.94 ± 0.01 |
| Imp. (%) | -0.29% | 4.88% | 12.13% | 11.57% | 10.80% | 0.19% | 2.85% |
| w/ TAMI | 58.57 ± 0.92 | 94.49 ± 0.21 | 85.54 ± 0.45 | 92.64 ± 0.06 | 90.71 ± 0.64 | 81.55 ± 0.10 | 97.93 ± 0.01 |
| Imp. (%) | 1.93% | 4.28% | 12.63% | 11.62% | 10.55% | 0.25% | 2.84% |
| DyGFormer | 88.14 ± 0.19 | 98.06 ± 0.01 | 89.86 ± 0.28 | 97.79 ± 0.01 | 94.20 ± 0.17 | 87.24 ± 0.35 | 98.83 ± 0.03 |
| w/ LTE | 88.72 ± 0.36 | 98.10 ± 0.01 | 89.97 ± 0.51 | 97.82 ± 0.02 | 94.94 ± 0.04 | 88.30 ± 0.28 | 98.93 ± 0.03 |
| Imp. (%) | 0.66% | 0.04% | 0.12% | 0.03% | 0.79% | 1.21% | 0.10% |
| w/ LHA | 89.41 ± 0.21 | 98.28 ± 0.03 | 90.36 ± 0.49 | 97.91 ± 0.01 | 94.24 ± 0.08 | 87.45 ± 0.01 | 98.87 ± 0.13 |
| Imp. (%) | 1.44% | 0.22% | 0.56% | 0.12% | 0.04% | 0.24% | 0.04% |
| w/ TAMI | 88.97 ± 0.41 | 98.30 ± 0.00 | 89.87 ± 0.06 | 97.85 ± 0.02 | 95.08 ± 0.13 | 88.08 ± 0.23 | 98.91 ± 0.00 |
| Imp. (%) | 0.94% | 0.24% | 0.01% | 0.06% | 0.93% | 0.96% | 0.08% |

| Methods | SE | UC | UT | UV | US | WK |
|---|---|---|---|---|---|---|
| GraphMixer | 91.73 ± 0.13 | 91.39 ± 0.05 | 62.26 ± 0.13 | 50.58 ± 0.45 | 50.72 ± 0.64 | 96.47 ± 0.13 |
| w/ LTE | 91.81 ± 0.11 | 92.07 ± 0.14 | 61.92 ± 0.37 | 50.76 ± 0.34 | 50.87 ± 0.21 | 96.58 ± 0.13 |
| Imp. (%) | 0.09% | 0.75% | -0.54% | 0.37% | 0.30% | 0.11% |
| w/ LHA | 91.91 ± 0.01 | 92.84 ± 0.13 | 63.05 ± 0.83 | 55.37 ± 0.22 | 50.99 ± 0.96 | 98.11 ± 0.01 |
| Imp. (%) | 0.20% | 1.59% | 1.28% | 9.47% | 0.53% | 1.70% |
| w/ TAMI | 93.03 ± 0.46 | 93.44 ± 0.29 | 62.40 ± 1.50 | 56.63 ± 0.36 | 51.44 ± 0.75 | 98.17 ± 0.01 |
| Imp. (%) | 1.44% | 2.24% | 0.22% | 11.96% | 1.42% | 1.76% |
| DyGFormer | 93.16 ± 0.08 | 94.40 ± 0.21 | 63.86 ± 0.35 | 56.16 ± 0.07 | 54.93 ± 0.53 | 98.54 ± 0.07 |
| w/ LTE | 93.21 ± 0.10 | 95.62 ± 0.05 | 64.94 ± 0.08 | 56.27 ± 0.16 | 54.06 ± 1.03 | 98.86 ± 0.01 |
| Imp. (%) | 0.05% | 1.29% | 1.69% | 0.20% | -1.58% | 0.32% |
| w/ LHA | 94.49 ± 0.01 | 94.62 ± 0.01 | 63.65 ± 0.18 | 56.20 ± 0.47 | 55.85 ± 0.20 | 98.63 ± 0.06 |
| Imp. (%) | 1.42% | 0.23% | -0.33% | 0.07% | 1.67% | 0.09% |
| DyGFormer | 93.17 ± 0.01 | 95.67 ± 0.09 | 64.25 ± 0.24 | 56.23 ± 0.17 | 55.08 ± 0.21 | 98.88 ± 0.07 |
| Imp. (%) | 0.01% | 1.35% | 0.61% | 0.12% | 0.27% | 0.35% |

Table 18: Inductive link prediction performance. AP of methods under various numbers of negative links during testing. **NEG=50** indicates that each positive link is evaluated against 50 negative links in the AP computation. For dataset names, please kindly refer to Section B.1.

| Methods | EN | LA | UC | UV | Methods | EN | LA | UC | UV |
|---|---|---|---|---|---|---|---|---|---|
| **NEG = 1** | | | | | | | | | |
| GraphMixer | 75.94 ± 0.21 | 82.06 ± 0.25 | 91.39 ± 0.05 | 50.58 ± 0.45 | DyGFormer | 89.86 ± 0.28 | 94.20 ± 0.17 | 94.40 ± 0.21 | 56.16 ± 0.07 |
| w/ TAMI | 85.54 ± 0.45 | 90.71 ± 0.64 | 93.44 ± 0.29 | 56.63 ± 0.36 | w/ TAMI | 89.87 ± 0.06 | 95.08 ± 0.13 | 95.67 ± 0.09 | 56.23 ± 0.17 |
| Imp. (%) | 12.63% | 10.55% | 2.24% | 11.96% | Imp. (%) | 0.01% | 0.93% | 1.35% | 0.12% |
| **NEG = 5** | | | | | | | | | |
| GraphMixer | 41.80 ± 0.35 | 58.37 ± 0.69 | 77.20 ± 0.05 | 17.38 ± 0.62 | DyGFormer | 66.88 ± 0.33 | 81.90 ± 0.52 | 85.86 ± 0.45 | 21.26 ± 0.30 |
| w/ TAMI | 57.22 ± 0.37 | 72.63 ± 1.95 | 81.75 ± 0.74 | 20.89 ± 0.21 | w/ TAMI | 67.31 ± 0.08 | 84.37 ± 0.59 | 88.02 ± 0.28 | 21.50 ± 0.47 |
| Imp. (%) | 36.91% | 24.43% | 5.90% | 20.20% | Imp. (%) | 0.64% | 3.01% | 2.51% | 1.13% |
| **NEG = 25** | | | | | | | | | |
| GraphMixer | 15.16 ± 0.33 | 33.93 ± 1.25 | 55.31 ± 0.21 | 4.14 ± 0.22 | DyGFormer | 34.66 ± 0.24 | 58.96 ± 1.15 | 71.89 ± 1.73 | 5.31 ± 0.13 |
| w/ TAMI | 24.19 ± 0.09 | 45.18 ± 4.36 | 60.48 ± 1.58 | 5.09 ± 0.08 | w/ TAMI | 35.85 ± 0.04 | 63.78 ± 2.10 | 75.52 ± 0.51 | 5.43 ± 0.11 |
| Imp. (%) | 59.58% | 33.18% | 9.35% | 22.97% | Imp. (%) | 3.42% | 8.18% | 5.06% | 2.26% |
| **NEG = 50** | | | | | | | | | |
| GraphMixer | 8.87 ± 0.21 | 24.85 ± 1.30 | 45.00 ± 0.51 | 2.15 ± 0.15 | DyGFormer | 22.53 ± 0.04 | 46.74 ± 1.36 | 62.61 ± 3.26 | 2.76 ± 0.08 |
| w/ TAMI | 14.50 ± 0.30 | 33.66 ± 4.48 | 49.54 ± 2.18 | 2.64 ± 0.05 | w/ TAMI | 23.72 ± 0.06 | 53.47 ± 1.61 | 67.75 ± 0.70 | 2.79 ± 0.05 |
| Imp. (%) | 63.42% | 35.45% | 10.08% | 22.84% | Imp. (%) | 5.28% | 14.39% | 8.21% | 1.09% |

Table 19: AP for inductive link prediction. The proposed LTE and LHA can be integrated into various types of TGNN. No LTE and TAMI results are reported for JODIE because it does not utilize the time encoding function.

| Method | EN | LA | UC | UV |
|---|---|---|---|---|
| TGN | 78.74 ± 1.53 | 80.05 ± 4.00 | 86.66 ± 1.83 | 54.13 ± 2.37 |
| TGAT | 67.63 ± 0.96 | 78.48 ± 0.07 | 79.11 ± 0.28 | 51.98 ± 0.14 |
| CAWN | 84.97 ± 0.21 | 89.48 ± 0.02 | 92.45 ± 0.06 | 49.29 ± 0.87 |
| JODIE | 80.67 ± 2.11 | 78.31 ± 3.82 | 79.27 ± 1.97 | 56.86 ± 0.06 |
| **w/ LTE** | | | | |
| TGN | 84.02 ± 1.63 | 89.33 ± 0.87 | 91.56 ± 1.41 | 57.92 ± 3.41 |
| Imp. (%) | 6.71% | 11.59% | 5.66% | 7.01% |
| TGAT | 77.70 ± 0.34 | 88.27 ± 0.05 | 92.61 ± 0.29 | 50.03 ± 0.31 |
| Imp. (%) | 14.89% | 12.47% | 17.07% | -3.75% |
| CAWN | 86.88 ± 0.13 | 91.14 ± 0.05 | 94.64 ± 0.08 | 48.54 ± 0.25 |
| Imp. (%) | 2.25% | 1.86% | 2.37% | -1.51% |
| **w/ LHA** | | | | |
| TGN | 85.97 ± 0.78 | 92.35 ± 0.44 | 89.55 ± 0.18 | 57.26 ± 3.58 |
| Imp. (%) | 9.18% | 15.37% | 3.34% | 5.79% |
| TGAT | 83.89 ± 0.01 | 90.26 ± 0.33 | 88.16 ± 0.13 | 54.42 ± 1.02 |
| Imp. (%) | 24.04% | 15.01% | 11.44% | 4.69% |
| CAWN | 86.24 ± 0.15 | 91.03 ± 0.06 | 92.74 ± 0.06 | 55.62 ± 1.66 |
| Imp. (%) | 1.49% | 1.73% | 0.31% | 12.84% |
| JODIE | 85.48 ± 0.24 | 91.71 ± 0.56 | 83.56 ± 0.08 | 57.64 ± 1.41 |
| Imp. (%) | 5.97% | 17.11% | 5.42% | 1.37% |
| **w/ TAMI** | | | | |
| TGN | 85.54 ± 0.45 | 90.71 ± 0.64 | 93.44 ± 0.29 | 57.63 ± 0.36 |
| Imp. (%) | 8.64% | 13.32% | 7.82% | 6.47% |
| TGAT | 84.53 ± 0.13 | 93.32 ± 0.20 | 94.25 ± 0.10 | 54.29 ± 0.30 |
| Imp. (%) | 24.98% | 18.91% | 19.15% | 4.44% |
| CAWN | 87.45 ± 0.23 | 92.61 ± 0.04 | 94.69 ± 0.11 | 64.33 ± 1.37 |
| Imp. (%) | 2.92% | 3.50% | 2.42% | 30.53% |

Table 20: Skewness of temporal differences before and after applying our LTE on the three TGB datasets tested.

| Datasets | tgbl-wiki | tgbl-review | tgbl-coin |
|---|---|---|---|
| Originally | 4.181 | 3.297 | 4.562 |
| with LTE | -0.307 | -0.524 | 0.308 |

Table 21: The standard deviations of five runs for results in Table 1 and Table 12.

| NSS | Methods | CP | CO | EN | FL | LA | MO | RE | SE | UC | UT | UV | US | WK |
|---|---|---|---|---|---|---|---|---|---|---|---|---|---|---|
| rnd | JODIE | 0.31 | 1.33 | 0.30 | 1.73 | 2.13 | 2.44 | 0.14 | 0.55 | 1.09 | 0.31 | 0.81 | 1.52 | 0.14 |
| | DyRep | 2.76 | 0.15 | 3.36 | 0.72 | 2.21 | 0.49 | 0.04 | 0.30 | 2.30 | 0.93 | 0.80 | 0.39 | 0.06 |
| | TGAT | 0.72 | 0.09 | 0.97 | 0.18 | 0.21 | 0.15 | 0.02 | 0.17 | 0.70 | 0.18 | 0.98 | 3.16 | 0.06 |
| | TGN | 2.34 | 0.56 | 1.11 | 0.14 | 3.97 | 1.60 | 0.06 | 0.17 | 1.04 | 1.37 | 2.17 | 0.58 | 0.06 |
| | CAWN | 2.34 | 0.28 | 0.09 | 0.01 | 0.06 | 0.25 | 0.01 | 0.09 | 0.06 | 0.12 | 0.10 | 0.48 | 0.03 |
| | EdgeBank | 0.00 | 0.00 | 0.00 | 0.00 | 0.00 | 0.00 | 0.00 | 0.00 | 0.00 | 0.00 | 0.00 | 0.00 | 0.00 |
| | TCL | 2.67 | 0.12 | 0.71 | 0.02 | 2.16 | 0.24 | 0.02 | 0.16 | 1.63 | 0.03 | 0.03 | 0.48 | 0.16 |
| | GraphMixer | 0.46 | 0.09 | 0.06 | 0.01 | 0.02 | 0.04 | 0.01 | 0.16 | 0.05 | 0.13 | 0.09 | 0.23 | 0.04 |
| | DyGFormer | 1.89 | 0.08 | 0.50 | 0.01 | 1.53 | 0.17 | 0.01 | 0.11 | 1.15 | 0.02 | 0.02 | 0.34 | 0.11 |
| | **w/ TAMI** | | | | | | | | | | | | | |
| | GraphMixer | 0.07 | 0.06 | 0.06 | 0.01 | 0.41 | 0.04 | 0.04 | 0.11 | 0.21 | 0.49 | 0.08 | 0.36 | 0.00 |
| | DyGFormer | 0.12 | 0.01 | 0.21 | 0.00 | 0.08 | 0.14 | 0.00 | 0.03 | 0.21 | 0.04 | 0.03 | 0.28 | 0.01 |
| hist | JODIE | 0.63 | 2.13 | 2.70 | 2.59 | 3.81 | 1.25 | 0.36 | 6.78 | 5.80 | 1.83 | 0.81 | 5.76 | 0.66 |
| | DyRep | 1.23 | 0.20 | 2.76 | 0.99 | 2.45 | 1.12 | 0.31 | 0.43 | 3.14 | 1.07 | 1.12 | 2.25 | 0.56 |
| | TGAT | 0.84 | 0.52 | 1.05 | 0.18 | 0.24 | 0.62 | 0.20 | 0.44 | 0.91 | 0.91 | 2.14 | 6.60 | 0.22 |
| | TGN | 3.07 | 2.35 | 1.76 | 1.64 | 4.64 | 1.93 | 0.61 | 0.56 | 2.12 | 5.51 | 3.93 | 7.57 | 0.33 |
| | CAWN | 2.77 | 0.49 | 0.36 | 0.97 | 0.43 | 0.95 | 0.45 | 0.38 | 0.43 | 0.38 | 0.04 | 8.23 | 1.67 |
| | EdgeBank | 0.00 | 0.00 | 0.00 | 0.00 | 0.00 | 0.00 | 0.00 | 0.00 | 0.00 | 0.00 | 0.00 | 0.00 | 0.00 |
| | TCL | 3.00 | 0.21 | 0.39 | 0.24 | 2.31 | 0.41 | 0.16 | 0.31 | 2.74 | 1.17 | 2.35 | 3.95 | 0.39 |
| | GraphMixer | 0.87 | 0.41 | 0.92 | 0.26 | 0.49 | 0.92 | 0.18 | 0.31 | 1.35 | 1.22 | 1.60 | 1.02 | 0.10 |
| | DyGFormer | 0.31 | 0.06 | 0.23 | 0.49 | 0.48 | 0.66 | 0.67 | 0.14 | 0.82 | 1.40 | 1.58 | 3.38 | 2.56 |
| | **w/ TAMI** | | | | | | | | | | | | | |
| | GraphMixer | 0.12 | 0.02 | 0.68 | 0.11 | 1.06 | 0.62 | 0.29 | 0.03 | 0.53 | 2.15 | 0.32 | 1.11 | 0.57 |
| | DyGFormer | 0.01 | 0.02 | 0.01 | 2.93 | 1.84 | 0.89 | 0.10 | 0.04 | 0.32 | 1.46 | 2.12 | 2.50 | 0.24 |
| ind | JODIE | 0.66 | 1.78 | 0.98 | 4.02 | 4.49 | 2.19 | 0.16 | 4.11 | 1.40 | 1.48 | 1.48 | 5.13 | 0.79 |
| | DyRep | 0.86 | 0.10 | 1.53 | 1.82 | 2.70 | 0.95 | 0.26 | 0.48 | 1.76 | 1.24 | 1.24 | 1.16 | 1.58 |
| | TGAT | 1.21 | 0.48 | 1.36 | 0.26 | 0.17 | 0.64 | 0.24 | 0.44 | 0.84 | 1.24 | 1.24 | 5.82 | 0.16 |
| | TGN | 2.87 | 3.28 | 2.72 | 2.72 | 5.98 | 2.91 | 0.24 | 0.56 | 0.71 | 6.01 | 6.01 | 6.47 | 0.44 |
| | CAWN | 1.00 | 0.27 | 0.58 | 1.52 | 0.77 | 0.94 | 0.24 | 0.27 | 0.48 | 0.67 | 0.67 | 8.52 | 2.62 |
| | EdgeBank | 0.00 | 0.00 | 0.00 | 0.00 | 0.00 | 0.00 | 0.00 | 0.00 | 0.00 | 0.00 | 0.00 | 0.00 | 0.00 |
| | TCL | 1.75 | 0.21 | 0.32 | 0.18 | 0.89 | 0.54 | 0.29 | 0.36 | 1.11 | 1.74 | 1.74 | 3.34 | 0.72 |
| | GraphMixer | 0.63 | 0.35 | 0.79 | 0.21 | 0.33 | 0.92 | 0.11 | 0.33 | 0.51 | 1.29 | 1.29 | 0.84 | 0.17 |
| | DyGFormer | 0.57 | 0.28 | 0.89 | 1.78 | 0.50 | 0.69 | 0.40 | 0.10 | 1.71 | 1.02 | 1.02 | 3.62 | 5.38 |
| | **w/ TAMI** | | | | | | | | | | | | | |
| | GraphMixer | 0.28 | 0.74 | 0.29 | 0.69 | 0.06 | 0.38 | 0.09 | 0.00 | 0.39 | 0.68 | 0.04 | 0.01 | 0.24 |
| | DyGFormer | 0.09 | 0.06 | 1.39 | 2.82 | 1.63 | 1.38 | 0.21 | 0.21 | 0.35 | 3.27 | 1.44 | 2.86 | 0.08 |

Table 22: The standard deviations of five runs for results in Table 3.

| Methods | CP | CO | EN | FL | LA | MO | RE | SE | UC | UT | UV | US | WK |
|---|---|---|---|---|---|---|---|---|---|---|---|---|---|
| GraphMixer | 0.46 | 0.09 | 0.06 | 0.01 | 0.02 | 0.04 | 0.01 | 0.16 | 0.05 | 0.13 | 0.09 | 0.23 | 0.04 |
| w/ LTE | 0.11 | 0.18 | 0.34 | 0.01 | 0.45 | 0.15 | 0.01 | 0.12 | 0.02 | 0.35 | 0.23 | 0.18 | 0.01 |
| w/ LHA | 0.60 | 0.05 | 0.04 | 0.03 | 0.07 | 0.19 | 0.02 | 0.06 | 0.18 | 0.19 | 0.17 | 0.33 | 0.00 |
| w/ TAMI | 0.07 | 0.06 | 0.06 | 0.01 | 0.41 | 0.04 | 0.04 | 0.11 | 0.21 | 0.29 | 0.08 | 0.36 | 0.00 |
| DyGFormer | 0.26 | 0.00 | 0.04 | 0.01 | 0.19 | 0.24 | 0.01 | 0.08 | 0.30 | 0.94 | 0.32 | 0.91 | 0.05 |
| w/ LTE | 0.07 | 0.01 | 0.06 | 0.00 | 0.00 | 0.11 | 0.01 | 0.01 | 0.22 | 0.01 | 0.45 | 0.69 | 0.02 |
| w/ LHA | 0.12 | 0.01 | 0.06 | 0.01 | 0.34 | 0.33 | 0.01 | 0.00 | 0.05 | 0.66 | 1.09 | 0.51 | 0.05 |
| w/ TAMI | 0.12 | 0.01 | 0.21 | 0.00 | 0.08 | 0.14 | 0.00 | 0.03 | 0.21 | 0.04 | 0.03 | 0.28 | 0.01 |

Table 23: The standard deviations of five runs for results in Table 4.

| Method | EN | LA | UC | UV | Method | EN | LA | UC | UV |
|---|---|---|---|---|---|---|---|---|---|
| **NEG = 1** | | | | | | | | | |
| GraphMixer | 0.06 | 0.02 | 0.05 | 0.09 | DyGFormer | 0.04 | 0.19 | 0.30 | 0.32 |
| w/ TAMI | 0.06 | 0.41 | 0.21 | 0.08 | w/ TAMI | 0.21 | 0.08 | 0.21 | 0.33 |
| **NEG = 5** | | | | | | | | | |
| GraphMixer | 0.39 | 0.23 | 0.09 | 0.16 | DyGFormer | 0.07 | 0.71 | 0.69 | 0.26 |
| w/ TAMI | 0.68 | 1.36 | 0.43 | 0.04 | w/ TAMI | 0.62 | 0.51 | 0.46 | 0.44 |
| **NEG = 25** | | | | | | | | | |
| GraphMixer | 0.20 | 0.53 | 0.27 | 0.05 | DyGFormer | 0.04 | 1.46 | 1.18 | 0.19 |
| w/ TAMI | 1.19 | 2.62 | 1.10 | 0.02 | w/ TAMI | 0.52 | 1.58 | 0.57 | 0.13 |
| **NEG = 50** | | | | | | | | | |
| GraphMixer | 0.16 | 0.52 | 0.08 | 0.04 | DyGFormer | 0.11 | 1.82 | 1.60 | 0.06 |
| w/ TAMI | 0.90 | 2.54 | 1.73 | 0.01 | w/ TAMI | 0.59 | 2.13 | 0.53 | 0.11 |

