# OpenReview forum: "TAMI: Taming Heterogeneity in Temporal Interactions for Temporal Graph Link Prediction"
_NeurIPS.cc/2025/Conference — NeurIPS 2025 poster_

### Official Review · Reviewer_LnpK · 2025-06-22

**Clarity:** 3
**Significance:** 2
**Originality:** 2
**Rating:** 4
**Confidence:** 5

**Summary:**

This paper proposes two enhancements to existing temporal link prediction methods: (1) applying a logarithmic transformation to the temporal difference $\Delta t$, and (2) storing the most recent $k$ embeddings for each edge instead of only the latest one. Experimental results demonstrate the effectiveness of the proposed approach on selected baselines.

**Questions:**

Please see the weakness part.

**Ethical Concerns:**

["NO or VERY MINOR ethics concerns only"]

**Final Justification:**

The rebuttal has addressed my concerns, so I decide to raise my score to borderline accept.

**Limitations:**

The limitations have been provided in Appendix C.3

**Paper Formatting Concerns:**

There are no paper formatting concerns.

**Quality:**

2

**Strengths And Weaknesses:**

**Strength**

1. The authors propose two enhancements that can be integrated into existing temporal link prediction methods.
2. Experimental results verify the effectiveness and efficiency of the proposed methods on the selected baselines and datasets.

**Weaknesse**

1. The technical contribution of this paper is limited, as the proposed modifications are trivial and their effects are expected. Specifically, the LTE method introduces a log transformation to the temporal differences—a well-known technique in machine learning to improve the training stability when the feature scales vary. As such, it is unsurprising that this modification improves model performance when the scales of temporal differences vary. Furthermore, the instability of temporal encoding methods is not a novel observation by the authors—it was previously discussed in GraphMixer [4]. Regarding the LHA module, storing more historical embeddings instead of just the most recent one to mitigate information loss is also an intuitive strategy. Similar ideas have been explored in prior work [3], and the reviewer does not find the proposed LHA design to offer significant novelty or technical depth.
2. The experimental evaluation lacks comprehensiveness and fails to convincingly support the claimed advantages of the proposed method..
    1. The newest baseline is published on NeurIPS 2023 (DyGFormer). There are lots of newer link prediction methods that are both more efficient and effective than DyGForemer [1,2,3]. The authors are supposed to report the results of TAMI on these methods, like [1], for a fair and thorough evaluation of the proposed TAMI method.
    2. The efficiency analysis is limited to two small datasets (i.e., UC and EN) and one computationally expensive baseline. To fully evaluate the efficiency of TAMI, the authors are encouraged to report results on larger datasets and more baselines.


[1] Lu X, Sun L, Zhu T, et al. Improving Temporal Link Prediction via Temporal Walk Matrix Projection[C]//The Thirty-eighth Annual Conference on Neural Information Processing Systems.

[2] Tian Y, Qi Y, Guo F. Freedyg: Frequency enhanced continuous-time dynamic graph model for link prediction[C]//The Twelfth International Conference on Learning Representations. 2024.

[3] Cheng K, Linzhi P, Ye J, et al. Co-Neighbor Encoding Schema: A Light-cost Structure Encoding Method for Dynamic Link Prediction[C]//Proceedings of the 30th ACM SIGKDD Conference on Knowledge Discovery and Data Mining. 2024: 421-432.

[4] Cong W, Zhang S, Kang J, et al. DO WE REALLY NEED COMPLICATED MODEL ARCHITECTURES FOR TEMPORAL NETWORKS?[C]//11th International Conference on Learning Representations, ICLR 2023. 2023.

---

> ### Author Rebuttal · Authors · 2025-07-31
>
> Thank you very much for the constructive feedback. Below we respond to the concerns raised.
>
> - **W1. The technical contribution of this paper is limited, as the proposed modifications are trivial and their effects are expected. Specifically, the LTE method introduces a log transformation to the temporal differences — a well-known technique in machine learning to improve the training stability when the feature scales vary. As such, it is unsurprising that this modification improves model performance when the scales of temporal differences vary**
>
> Thanks for the thoughtful comment. We believe that **the reviewer's assessment that our modules' effects were as expected, and the performance improvement was unsurprising, confirms the intuitive and effective nature of our design**.
>
> We emphasize that our design is novel and our contributions are substantial. Specifically, we are the first to identify the heterogeneity in temporal interactions and investigate its impact on the performance of TGNNs thoroughly. This heterogeneity leads to two problems in temporal link prediction for existing TGNNs, i.e., ineffective temporal encoding and forgetting of past interactions for a pair of nodes that interact intermittently. To cope with these problems, we propose two novel modules, namely, LTE and LHA, respectively. We then develop an end-to-end framework named TAMI to integrate LTE and LHA into existing TGNNs to boost their performance in temporal link prediction. Extensive experiment results on 13 classic datasets and 3 large TGB datasets demonstrate the effectiveness of TAMI.
>
> In particular, **while the logarithmic transformation is a well-known technique in statistics and machine learning, it remains unknown why and where we need to apply it to TGNNs**. LTE is motivated by our observation that temporal differences often follow a highly right-skewed distribution, but existing TGNNs overlook this skewness. Since their time encoding functions are directly applied to the raw temporal differences, their resulting models often struggle with the prediction of infrequent interactions, as shown in Figure 3(b). LTE addresses these issues by transforming a skewed distribution of temporal differences into a more balanced one, thereby effectively improving the link prediction accuracy.
>
> - **W1 (cont'd) The instability of temporal encoding methods was previously discussed in GraphMixer.**
>
> Thanks for the comment. Our work is **orthogonal** to GraphMixer. GraphMixer finds that the instability of time encoding functions is due to the fact that the gradients of their trainable parameters scale proportionally with the input temporal differences. Thus, they propose to fix these parameters during training. In contrast, in TAMI, we observe a new problem that **the distribution of the input temporal differences is often right-skewed**, and its negative impact on temporal link prediction. Based on this observation, we propose a novel time encoding function, LTE, to directly mitigate the skewness in the temporal differences to improve the link prediction performance. Note that LTE can be used for both when the frequency parameters need to be learned or fixed.
>
> As such, LTE can be integrated into GraphMixer to boost its performance. As shown in Figure 3(b), together with LTE, GraphMixer's performance improves substantially, especially for node pairs whose interaction intervals are long. Furthermore, Tables 3 and 16 in our manuscript show that LTE, used with the fixed time encoding function in GraphMixer, consistently improves GraphMixer's performance under both transductive and inductive settings.
>
> - **W1 (cont'd) LHA stores more historical embeddings instead of just the most recent one to mitigate information loss. Similar ideas have been explored in prior work [3].**
>
> Thanks for the comment. Again, our work is **orthogonal** to [3]. Our LHA module maintains the historical interactions using edge embeddings for each target node pair, in addition to the node embeddings learned by the underlying TGNN, such as [3], which retains each node's recent historical interactions to learn its temporal node embeddings.
>
> Suppose we are to predict a link between node $u$ and node $v$. [3] first computes the temporal node embeddings of $u$ and $v$ using $u$'s recent interactions and $v$'s recent interactions, respectively. However, it is possible that $u$ and $v$ do not interact recently. Hence, their node embeddings may not capture any of their interaction history and thus be less effective in link prediction. In contrast, LHA explicitly encodes the past interactions between $u$ and $v$ into a dedicated historical edge embedding and directly leverages this edge embedding, together with their node embeddings, for predicting their future link.
>
> [3] Cheng K, Linzhi P, Ye J, et al. Co-Neighbor Encoding Schema: A Light-cost Structure Encoding Method for Dynamic Link Prediction. ACM KDD 2024.
>
> - **W2. The experimental evaluation lacks comprehensiveness and fails to convincingly support the claimed advantages of the proposed method**
>
> Thanks for the comment. Our experimental evaluation is comprehensive and extensive, as it is done with various underlying TGNNs on 13 classic datasets and 3 large TGB datasets under both inductive and transductive settings with three negative sampling strategies.
>
> Furthermore, the effectiveness of LTE and LHA is demonstrated separately. The effectiveness of LTE in mitigating the skewness in temporal differences for improved link prediction accuracy is well supported by Figure 3(b) and Tables 3, 13, 16, and 18 in our manuscript. In addition, the effectiveness of LHA in addressing the forgetting issue of historical interactions between target node pairs for improved link prediction accuracy is well demonstrated by Figures 5, 8, and 9, and Tables 3, 13, 16, and 18.
>
> - **W2 (cont'd) Authors are supposed to report the results of TAMI on newer methods, like [1], for a fair and thorough evaluation of the proposed TAMI method**
>
> Thanks for the comment. We do not aim to propose yet another TGNN architecture. Instead, we propose TAMI to be used for existing TGNNs to further improve their performance by addressing the heterogeneity of temporal interactions that is overlooked in their learning process. We demonstrate the effectiveness of TAMI through extensive experiments. For example, Tables 1, 2, and 5 in our manuscript show that TAMI consistently improves the performance of various TGNNs over their vanilla counterparts.
>
> To provide a more thorough evaluation, we have conducted extra experiments on TAMI with TPNet [1]. As shown in Table 1 below, TAMI indeed improves the performance of TPNet.
>
> **Table 1: Transductive AP of TPNet with and without TAMI.**
> | Method \ Datasets|USLegis (US) |Enron (EN) |Uci (UC) |CanParl (CP) |
> |-|-|-|-|-|
> |TPNet|80.58|92.90|97.35|90.28|
> |**TPNet+TAMI**|**94.15**|**93.87**|**97.61**|**91.03**|
>
> [1] Lu X, Sun L, Zhu T, et al. Improving Temporal Link Prediction via Temporal Walk Matrix Projection. NeurIPS 2024
>
> - **W2 (cont'd) The efficiency analysis is limited to two small datasets (i.e., UC and EN) and one computationally expensive baseline**
>
> Thanks for the comment. In Section 4.6, we do not aim to evaluate the computational efficiency of TAMI, but rather to empirically demonstrate that TAMI can improve the training efficiency of underlying TGNNs. As shown in Figure 6, TAMI enables **DyGFormer, GraphMixer, and TGAT** to converge **faster** than their vanilla counterparts, while achieving higher validation average precision with fewer training epochs. We have also conducted similar experiments for three other baselines, i.e., **TGN, JODIE, and CAWN**. Results in Table 2 below show that TAMI indeed improves their training efficiency. The overall trend of training three new baselines is similar to our observation with Figure 6.
>
> **Table 2: Results on the UC (top) and EN (bottom) dataset.**
> |Method|Val AP after 5 epochs|Final Val AP|Total epochs|
> |-|-|-|-|
> |TGN|85.28|88.32|41|
> |**+TAMI**|**94.14**|**94.43**|**36**|
> |JODIE|83.20|84.05|39|
> |**+TAMI**|**89.12**|**89.36**|**24**|
> |CAWN|93.22|93.53|**42**|
> |**+TAMI**|**95.18**|**95.43**|48|
>
> |Method|Val AP after 5 epochs|Final Val AP|Total epochs|
> |-|-|-|-|
> |TGN |80.61|87.01|70|
> |**+TAMI**|**90.21**|**91.10**|**43**|
> |JODIE|77.24|85.77|82|
> |**+TAMI**|**88.71**|**89.52**|**28**|
> |CAWN|88.79|90.10|36|
> |**+TAMI**|**90.19**|**91.18**|**34**|
>
> In addition, in Figure 7, we demonstrate that TAMI enables existing TGNNs, such as DyGFormer, to compute temporal node embeddings using **fewer** historical interactions to achieve comparable or even better performance. As a result, they consume much less GPU memory during training, require less training time per epoch, and converge faster with fewer epochs. We here use DyGFormer as an example because it is the top performer in all baselines. Together with TAMI, this computationally expensive baseline becomes much more efficient while achieving comparable or even better performance, with less GPU memory usage (up to 53.22%) and less training time per epoch (up to 40.54%).
>
> We have conducted additional experiments on the large-scale dataset, **UNVote (UV)**, which contains over one million edges, using DyGFormer and CAWN. Results are shown in Table 3 below. Again, TAMI allows them to use significantly fewer historical interactions to achieve comparable or even better performance, thereby reducing both GPU memory usage and total training time.
>
> **Table 3: Results on the UV dataset. The number in the parenthesis indicates the number of historical interactions used to compute the node embeddings.**
> |Method|GPU mem (MiB)|Train time per epoch (second) |Epochs|Test AP|
> |-|-|-|-|-|
> |DyGFormer(128)|4069|490|67|55.88|
> |DyGFormer(32)|2415|341|60|55.15|
> |**DyGFormer(32)+TAMI**|**2435**|**352**|**23**|**59.09**|
> |CAWN(64)|7279|2442|52|52.84|
> |CAWN(4)|2691|330|26|51.69|
> |**CAWN(4)+TAMI**|**2741**|**347**|**23**|**58.47**|

---

> > ### Author Response · Authors · 2025-08-04
> >
> > Dear Reviewer,
> >
> > Thank you again for recognizing the effectiveness and efficiency of our proposed methods.
> >
> > We have provided detailed explanations and additional experiment results in response to your concerns.
> >
> > Based on your suggestions, we will add the corresponding experiments and discussions to our revised manuscript.
> >
> > As the discussion period is halfway through, we would greatly appreciate your feedback on these responses to confirm that they fully meet your expectations. Please kindly let us know if you have any further questions. We will be very happy to answer.
> >
> > Sincerely,
> >
> > Authors

---

> > > ### Comment · Reviewer_LnpK · 2025-08-04
> > >
> > > Thanks for the detailed response. It has addressed my concerns, and I decide to raise my score to borderline accept.

---

> > > > ### Author Response · Authors · 2025-08-04
> > > >
> > > > Thank you very much for raising the score and the time spent on reviewing our work.
> > > >
> > > > Sincerely,
> > > >
> > > > Authors

---

### Official Review · Reviewer_PJpx · 2025-06-28

**Clarity:** 3
**Significance:** 2
**Originality:** 3
**Rating:** 4
**Confidence:** 4

**Summary:**

The paper addresses the challenge of temporal graph link prediction by identifying heterogeneity in interaction patterns, such as skewed interaction frequencies and irregular time intervals, which hinder effective temporal information encoding and cause forgetting of past interactions. It proposes TAMI, a novel framework with two components: Log Time Encoding, which balances skewed time intervals using logarithmic transformation, and Link History Aggregation, which prevents forgetting historical interactions for node pairs.TAMI integrates seamlessly with existing temporal graph neural networks, enhancing their performance. Extensive experiments on 13 classic datasets and 3 TGB datasets demonstrate that TAMI significantly improves link prediction accuracy and training efficiency.

**Questions:**

In Section 4.6, the authors state that “UC and EN datasets require only 2 and 32 historical interactions for attention calculation,” indicating that the choice of the number of historical interactions (k) varies across datasets. Is this variation related to dataset characteristics such as sparsity, scale, or interaction depth?

**Ethical Concerns:**

["NO or VERY MINOR ethics concerns only"]

**Limitations:**

The authors’ contributions focus on improving temporal encoding and information aggregation, offering a novel perspective and effectively enhancing existing models. However, the approach lacks methodological innovation, resulting in somewhat limited overall contributions.

**Paper Formatting Concerns:**

No major formatting issues.

**Quality:**

2

**Strengths And Weaknesses:**

Strengths
The paper is well-organized, providing a clear and comprehensive overview of Temporal Graph Link Prediction. It features precise formula definitions, well-formulated problem hypotheses, rigorous mathematical proofs, and extensive comparative and ablation experiments. The narrative is logical and coherent, ensuring a robust structure. Key strengths include being the first to identify heterogeneity in continuous-time temporal graph interactions and its impact on link prediction performance. The proposed TAMI framework includes Log Time Encoding, which rescales time differences via logarithmic transformation, and Link History Aggregation, which preserves historical interactions for target node pairs. Existing temporal graph neural networks can be seamlessly integrated into TAMI, enhancing their effectiveness.

Weaknesses
In Figure 4, when describing historical interactions between node pairs, node v5 appears twice as an interaction node for node u, which may be a typographical error and warrants author clarification. The paper does not discuss limitations or shortcomings of the proposed method. Although the authors conducted thorough experiments, they did not clarify whether the hyperparameter in Equation (4), used to measure the forgetting rate of historical interactions, varies across different datasets; if so, further explanation is needed.

---

> ### Author Rebuttal · Authors · 2025-07-31
>
> We thank the reviewer for recognizing our paper with **precise formula definitions, well-formulated problem hypotheses, rigorous mathematical proofs, and extensive comparative and ablation experiments** as well as **being the first to identify heterogeneity in continuous-time temporal graph interactions and its impact on link prediction performance**. Below, we respond to the concerns raised.
>
> - **W1. In Figure 4, when describing historical interactions between node pairs, node v5 appears twice as an interaction node for node u, which may be a typographical error and warrants author clarification**
>
> Thanks for pointing this out. We will update the figure to use two links, instead of two nodes, to represent the two interactions between node $u$ and node $v_5$ for better clarity.
>
> As you may have noticed, our intention behind Figure 4 was to show that a target node may engage in frequent interactions with a small set of nodes, dominating its recent interaction history. For example, a student may have hamburgers on both Monday and Tuesday, resulting in two distinct links between the "student" node and the "hamburger" node, each associated with a different timestamp. In this case, node $u$ frequently interacts with node $v_5$ at timestamps $t_3$ and $t_4$. As a result, historical interactions between the target node pair, i.e., node $u$ and node $v$, may be forgotten in their latest temporal node embeddings, thereby leading to degraded link prediction accuracy.
>
> - **W1 (cont'd). The paper does not discuss limitations or shortcomings of the proposed method**
>
> Thanks for the comment. We would like to point out that the limitations are discussed in Appendix C.3.
>
> - **W1 (cont'd). Although the authors conducted thorough experiments, they did not clarify whether the hyperparameter in Equation (4), used to measure the forgetting rate of historical interactions, varies across different datasets; if so, further explanation is needed.**
>
> Thanks for the comment. We would also like to point out that the setting of $\gamma$ in Equation (4) is mentioned in the section Implementation Details in Appendix B.3. We set $\gamma = 0.9$ for most experiments. In addition, we examine the performance of underlying TGNNs with different values of $\gamma$ and present the results in Table 14 under the section Analysis of the Hyperparameter $\gamma$ in LHA in Appendix C.4.
>
> - **Q1. In Section 4.6, the authors state that "UC and EN datasets require only 2 and 32 historical interactions for attention calculation", indicating that the choice of the number of historical interactions (k) varies across datasets. Is this variation related to dataset characteristics such as sparsity, scale, or interaction depth?**
>
> Thanks for the comment. We would like to clarify that the number of historical interactions discussed here is not the same as the number of **historical edge embeddings ($k$)** stored and used in our LHA module. **The value of $k$ is set to 1 and remains fixed in the experiments.** Existing TGNNs (e.g., GraphMixer and DyGformer) utilize historical interactions **of a node** to compute its temporal node embedding. However, in addition to those node embeddings, TAMI stores and leverages $k$ historical edge embeddings in LHA to ensure that the mutual interaction history between a target node pair is preserved when predicting their future links. Figure 7 in Section 4.6 demonstrates that, when integrated with TAMI, the underlying TGNN (e.g., DyGformer) can compute temporal node embeddings based on significantly fewer historical interactions compared with its vanilla counterpart, i.e., from 256 to 32 on the **EN** dataset and from 32 to 2 on the **UC** dataset, to achieve comparable or even better performance.
>
> - **Limitations: The authors’ contributions focus on improving temporal encoding and information aggregation, offering a novel perspective and effectively enhancing existing models. However, the approach lacks methodological innovation, resulting in somewhat limited overall contributions**
>
> Thanks for the thoughtful comment. We would like to point out that we are the first to identify the presence of heterogeneity in temporal interactions and investigate its impact on the performance of TGNNs thoroughly. This heterogeneity leads to two problems in temporal link prediction for existing TGNNs, i.e., ineffective temporal encoding and forgetting of past interactions for a pair of nodes that interact intermittently. To cope with these problems, we propose **two novel modules, namely, LTE and LHA**, respectively. We then develop **an end-to-end framework named TAMI** to integrate LTE and LHA with various types of existing TGNNs in order to boost their performance in temporal link prediction. In other words, we do not propose yet another TGNN, but we propose TAMI to be used as an add-on to existing TGNNs for improved temporal link prediction. Comprehensive and extensive experiment results on 13 classic datasets and 3 large TGB datasets with three negative sampling strategies under both inductive and transductive settings demonstrate the effectiveness of TAMI.
>
> We hope this clarifies the motivation and the contributions of our work, and we sincerely appreciate the reviewer’s comment.

---

> > ### Author Response · Authors · 2025-08-04
> >
> > Dear Reviewer,
> >
> > Thank you again for recognizing that we provide a novel perspective in our design and the comprehensiveness of our experiments.
> >
> > We have provided detailed explanations in response to your concerns.
> >
> > Based on your suggestions, we will update Figure 4 and include the discussions in our revised manuscript.
> >
> > As the discussion period is halfway through, we would greatly appreciate your feedback on these responses to confirm that they fully meet your expectations. Please kindly let us know if you have any further questions. We will be very happy to answer.
> >
> > Sincerely,
> >
> > Authors

---

> > > ### Author Response · Authors · 2025-08-08
> > >
> > > Dear Reviewer,
> > >
> > > Thank you very much for reviewing our paper. We hope our responses have addressed your concerns, and we sincerely appreciate that you can recommend accepting our paper. Please also let us know if you have any further questions, we will be very happy to answer.
> > >
> > > Best Regards,
> > >
> > > Authors

---

### Official Review · Reviewer_yD2r · 2025-06-28

**Clarity:** 3
**Significance:** 3
**Originality:** 3
**Rating:** 5
**Confidence:** 4

**Summary:**

This manuscript focuses on the problem of link prediction over temporal graphs. The authors first identify the observation that node interactions in temporal graphs are highly skewed in terms of frequency and interval, which negatively impacts the performance of existing temporal link predication methods. To address this issue, a light TAMI framework is proposed, consisting of two components LTE and LHA. LTE alleviates the skewness in node interactions, while LHA addresses the forgetting of historical interactions. Extensive experiments demonstrate that TAMI significantly enhances the performance of TGNNs on link prediction, while also accelerating convergence and reducing GPU memory usage.

**Questions:**

I hope the authors could provide a clear response to the weaknesses mentioned above, especially point (O1) and (O2).

**Ethical Concerns:**

["NO or VERY MINOR ethics concerns only"]

**Final Justification:**

Thank you for the author's submission. Appreciate the authors’ efforts and contributions. The final rating reflects a comprehensive evaluation of the paper’s strengths and limitations.

**Quality:**

3

**Strengths And Weaknesses:**

This manuscript has the following strengths.

(S1) The paper is well organized, with clear motivation for the LTE and LHA, effectively supported by Figure 3 and Figure 5, respectively.

(S2) The TAMI framework is light-weight and can be easily implemented and incorporated with various backbone TGNNs.

(S3) TAMI demonstrates effectiveness for both transductive and inductive link prediction.

Nevertheless, there are also several weaknesses in this manuscript that should be further improved.

(O1) In Section 3.3, the authors state that LHA uses most-recent aggregation. This effectively renders the parameter k redundant, as only a single historical interaction is considered regardless the value of k. Intuitively, using a larger k should enhance the performance of TAMI by incorporating more historical context. Therefore, additional experiments should be conducted with varying values of k, as well as alternative aggregation functions such as sum, which would make practical use of multiple historical interactions.

(O2) Some experimental results require more detailed interpretation. As shown in Table 11, datasets CO, LA, and SE exhibit higher skewness. However, Table 1 shows that the performance improvement obtained from TAMI on these datasets are relatively modest. Notably, the SE dataset, despite having a skewness over 140, shows less than 2% performance improvement, which is significantly lower than the improvements observed on other datasets. This finding appears to contradict the motivation that TAMI should be especially effective on highly skewed datasets. The authors should provide a deeper explanation for this discrepancy. Besides, the skewness of TGB generated datasets should also be reported.

(O3) Similarly, in Figure 5, the performance improvement achieved by LHA on exclusive nodes for datasets EN and UC is smaller than improvement achieved on isolated and mutual nodes. Intuitively, one would expect LHA to yield the most significant improvement on exclusive nodes. The authors should provide additional insights or analysis to clarify why LHA appears less effective on exclusive nodes in EN and UC compared to the other two datasets.

---

> ### Author Rebuttal · Authors · 2025-07-31
>
> We thank the reviewer for recognizing our work with **clear motivation** that is **effectively supported**, our framework as **light-weight** and **easily implemented**. Below we respond to the concerns raised.
>
> - **(O1) In Section 3.3, the authors state that LHA uses most-recent aggregation. This effectively renders the parameter k redundant, as only a single historical interaction is considered regardless the value of k. Intuitively, using a larger k should enhance the performance of TAMI by incorporating more historical context. Therefore, additional experiments should be conducted with varying values of k, as well as alternative aggregation functions such as sum, which would make practical use of multiple historical interactions.**
>
> Thanks for the comment. We respectfully disagree that only a single historical interaction is considered. We use an exponentially weighted moving average as in Equation (4) of the manuscript to compute the dedicated edge embedding for a given target pair of nodes. Thus, even if $k=1$, LHA considers all the interaction history between the pair of nodes. Varying $k$ values only controls how much freshness of information for the pair of nodes we maintain with a higher weight in the exponentially weighted moving average.
>
> In Table 1 below, we show the results of additional experiments to evaluate the performance of TAMI with different $k$ values and aggregation functions. Overall, the performance of different variants of TAMI is better than the vanilla version (without TAMI), indicating the effectiveness of our design. It is also important to note that larger $k$ value does not necessarily lead to better performance. This is expected as different datasets may have different levels of dependency on the interaction history for temporal link prediction. In addition, TAMI is not sensitive to the choice of aggregation functions.
>
> **Table 1: Test AP of DyGFormer on the EN and CP datasets with different $k$ values and aggregation strategies.**
> | Method | EN | CP |
> |-|-|-|
> | DyGFormer (Vanilla) | 92.46 | 97.91 |
> | w/ TAMI             |       |       |
> | Most-recent         | 92.66 | 98.67 |
> | Mean ($k=1$)        | 92.66 | 98.67 |
> | Mean ($k=2$)        | 92.60 | 98.76 |
> | **Mean ($k=3$)**        | 92.56 | **98.77** |
> | Max ($k=1$)         | 92.66 | 98.67 |
> | **Max ($k=2$)**         | **92.84** | 98.74 |
> | Max ($k=3$)         | 92.60 | 98.74 |
>
> - **(O2) Some experimental results require more detailed interpretation. As shown in Table 11, datasets CO, LA, and SE exhibit higher skewness. However, Table 1 shows that the performance improvement obtained from TAMI on these datasets are relatively modest. Notably, the SE dataset, despite having a skewness over 140, shows less than 2\% performance improvement, which is significantly lower than the improvements observed on other datasets. This finding appears to contradict the motivation that TAMI should be especially effective on highly skewed datasets. The authors should provide a deeper explanation for this discrepancy.**
>
> Thank you very much for this insightful comment. We respectfully argue that the performance gains achieved by TAMI on the **CO** and **LA** datasets are satisfactory. On the **LA** dataset, GraphMixer with TAMI improves by up to 16.64%, while DyGFormer with TAMI improves by up to 2.24% among all the three negative sampling strategies. Likewise, on the **CO** dataset, GraphMixer with TAMI improves by up to 5.77%, while DyGFormer with TAMI improves by up to 3.93%. These results all demonstrate the effectiveness of TAMI.
>
> For **SE (SocialEvo)**, we have looked into the dataset carefully and found that the high skewness is caused by a two-mode distribution. As shown in Table 2 below, one mode takes almost all the temporal differences while the other mode consists of only few ones (0.002%). There are no values between the two modes. This is due to the nature of the dataset. It is a mobile phone proximity network that captures daily interactions among students living in a dormitory over eight months. Temporal edges are formed between two students when they interact with each other via mobile phones. Hence, most interactions occur daily, but there are a few interactions that take place after more than two months, probably summer breaks, leading to ultra large temporal differences compared with the others. Even in this extreme case, TAMI can still bring an improvement by balancing the distribution, again showing the effectiveness of our design.
>
> We hope this clarifies and thanks again for the reviewer's meticulous review.
>
> **Table 2: The distribution of temporal differences on the SE dataset exhibits a bimodal pattern, with 99.998\% of the temporal differences being less than 19.67 days.**
> | Temporal differences |&nbsp; [0.0, 19.67) days |&nbsp; [19.67, 75.23) days |&nbsp; $\ge$ 75.23 days |
> |-|-|-|-|
> | Ratio  | &nbsp;  99.998\% | &nbsp; 0 | &nbsp; 0.002\%  |
>
> - **(O2) (cont'd). Besides, the skewness of TGB generated datasets should also be reported.**
>
> Thanks for the suggestion. We report the original skewness of temporal differences in the TGB datasets and the skewness after applying our LTE in Table 3 below. Results show that our LTE effectively reduces the skewness in the distribution of the temporal differences.
>
> **Table 3: Skewness of temporal differences before and after applying our LTE on the three TGB datasets tested**
> | Datasets    | tgbl-wiki | tgbl-review | tgbl-coin |
> |-------------|-----------|-------------|-----------|
> | Originally  | 4.181     | 3.297       | 4.562     |
> | with LTE    | -0.307    | -0.524      | 0.308     |
>
> - **(O3) Similarly, in Figure 5, the performance improvement achieved by LHA on exclusive nodes for datasets EN and UC is smaller than improvement achieved on isolated and mutual nodes. Intuitively, one would expect LHA to yield the most significant improvement on exclusive nodes. The authors should provide additional insights or analysis to clarify why LHA appears less effective on exclusive nodes in EN and UC compared to the other two datasets.**
>
> Thanks for the comment. The performance improvement achieved by LHA on exclusive nodes is smaller than that on isolated and mutual nodes in the **EN** and **UC** datasets. This is because most exclusive target node pairs have no interaction prior to prediction, resulting in empty LHA memory at prediction time, as shown in Table 4 below. For example, 89.67% of the exclusive target node pairs in the **UC** dataset have not interacted with each other during training. This is possible as the edges in the **UC** dataset are formed by messages sent between students on some social media. Many students may only interact once if they are not familiar with each other, which is used for their link prediction in testing. Without historical information stored, LHA cannot contribute to these predictions, leading to limited or no performance gains. However, all isolated and mutual node pairs have interacted at least once during training, allowing LHA to store and utilize historical interactions between target node pairs and thus improve performance.
>
> The case for exclusive node pairs from the **UC** dataset can also be treated as a special case of the cold start for LHA, which is studied in Appendix C.6 in the manuscript. As shown in Figure 9 in the manuscript, if we have more interaction events between target node pairs, the performance gain will be larger.
>
> **Table 4: Proportion of node pairs that *do not* have any interaction in training set**
> | Dataset | EN     | UC     | LA     | UV  |
> |-|-|-|-|-|
> | Exclusive node pairs | 41.55% | 89.67% | 37.31% | 0% |
> | Isolated node pairs  | 0%    | 0%    | 0%    | 0% |
> | Mutual node pairs    | 0%    | 0%    | 0%    | 0% |

---

> > ### Author Response · Authors · 2025-08-04
> >
> > Dear Reviewer,
> >
> > Thank you again for recognizing our work with clear motivation, our proposed framework as light-weight, and our experiments as comprehensive.
> >
> > We have provided detailed explanations and additional experiment results in response to your questions.
> >
> > Based on your suggestions, we will add the corresponding experiments and discussions to our revised manuscript.
> >
> > As the discussion period is halfway through, we would greatly appreciate your feedback on these responses to confirm that they fully meet your expectations. Please kindly let us know if you have any further questions. We will be very happy to answer.
> >
> > Sincerely,
> >
> > Authors

---

> > ### Author Response · Authors · 2025-08-08
> >
> > Dear Reviewer,
> >
> > Thank you very much for reviewing our paper. We hope our responses have addressed your concerns, and we sincerely appreciate that you can recommend accepting our paper. Please also let us know if you have any further questions, we will be very happy to answer.
> >
> > Best Regards,
> >
> > Authors

---

### Official Review · Reviewer_btob · 2025-07-03

**Clarity:** 4
**Significance:** 3
**Originality:** 3
**Rating:** 5
**Confidence:** 3

**Summary:**

This paper addresses limitations in existing time encoding methods for dynamic graphs by analyzing the statistical distribution of temporal interactions, showing that they follow a Pareto distribution. The authors argue that commonly used sinusoidal time encoding functions are biased towards frequent interactions and fail to capture long-range, infrequent ones effectively. Furthermore, they identify a limitation in the practice of using only the most recent $m$ interactions for node embeddings, which tends to neglect important historical context between node pairs. To address these issues, the paper introduces two key contributions: (1) Log Time Encoding (LTE), which applies a logarithmic transformation to rescale time differences, making temporal patterns easier to learn; and (2) Link History Aggregation (LHA), which preserves a fixed number of most recent interactions per target node pair to ensure rare but relevant past links are retained during link prediction. These design choices are theoretically motivated and empirically evaluated to demonstrate improved performance.

**Questions:**

1. [1] has found simple linear time encoding functions to be more effective than the sinusoidal function used in your work. Can the linear time encoding function benefit from your module?
2. LHA module maintains historical interactions for a target node pair that may be missed when performing temporal node message passing. Why does this approach beat RNN-like methods (e.g., TGN) that can also track very long distance interactions? How does this method compare to DyG2Vec [2] which also uses that past $k$ interactions for a target node pair to build temporal embeddings?
3. Does the choice of aggregation function for LHA affect performance?

[1] Chung et al. Between Linear and Sinusoidal: Rethinking the Time Encoder in Dynamic Graph Learning. TMLR

[2] Alomrani et al. DyG2Vec: Efficient Representation Learning for Dynamic Graphs. TMLR

**Ethical Concerns:**

["NO or VERY MINOR ethics concerns only"]

**Final Justification:**

The authors have conducted all the ablation and analysis experiments that I asked for.

**Limitations:**

yes

**Paper Formatting Concerns:**

Possible typo in figure 4: According to your notation. $t_5$ should be the oldest timestep while $t_1$ is the latest timestep but the figure suggests otherwise

**Quality:**

4

**Strengths And Weaknesses:**

Strengths:
1. Extensive experimental setup across several dynamic graph models (e.g., DyGFormer, TGN, etc), datasets, and settings (inductive, transductive).
2. The proposed LTE module can be easily integrated into any dynamic graph encoder. Experiments show that it almost never hurts performance.
3. Improvement is more pronounces when increasing the number of negative edges.

Weaknesses:
1. No experiments on other dynamic graph tasks (e.g., dynamic node classification).
2. Less improvement in the inductive setting and for balanced datasets where the temporal differences roughly follow a normal distribution.
3. The LTE and LHA modules have minor improvements for some models (e.g., DyGFormer). No provided intuition as to why this occurs.

---

> ### Author Rebuttal · Authors · 2025-07-31
>
> We thank the reviewer for recognizing our design choices as **theoretically motivated** and our experiments as **extensive**. Below we respond to the concerns raised.
>
> - **W1. No experiments on other dynamic graph tasks (e.g., dynamic node classification)**
>
> Thanks for the comment. We would like to clarify that our TAMI framework is specifically designed for the temporal link prediction task, which is the main downstream task in TGNNs. We observe that temporal interactions are heterogeneous, e.g., a few node pairs can make most interaction events, and interaction events happen at varying intervals. This can significantly degrade the link prediction performance of existing TGNNs, especially when predicting the links for infrequently interacting node pairs. Thus, we develop the TAMI framework to mitigate the heterogeneity issue in temporal interactions and in turn improve the link prediction accuracy. While it is beyond the scope of this work to extend TAMI to other dynamic graph tasks, such as dynamic node classification, we believe it is a promising direction for our future work.
>
> - **W2. Less improvement in the inductive setting and for balanced datasets where the temporal differences roughly follow a normal distribution.**
>
> Thanks for the thoughtful comment. For the inductive setting, it is similar to the case discussed in the section Cold Start of LHA in Inference in Appendix C.6. Specifically, the target node pairs in testing may not be observed during training. Hence, LHA's memory may need to be constructed from scratch during the test phase. At the beginning of the test phase, LHA contains no historical information for some node pairs, limiting its effectiveness and leading to smaller performance gains. However, as the test phase progresses and more interaction events are accumulated, LHA becomes more and more effective, resulting in higher performance gains.
>
> In addition, as shown in Table 11 in Appendix B.5, most of the datasets exhibit skewness (or heterogeneity in temporal interactions). TAMI is designed to tackle the problems associated with this skewness. Hence, when integrated into TAMI, the performance of underlying TGNNs improves substantially for most cases. For few balanced datasets, the performance gain from TAMI may not be as significant as the one with skewed datasets, but TAMI still enables underlying TGNNs to achieve better performance than their vanilla counterparts. This is somewhat expected because the skewness is mild in the balanced datasets. Nonetheless, many real-world datasets exhibit high skewness.
>
> - **W3. The LTE and LHA modules have minor improvements for some models (e.g., DyGFormer). No provided intuition as to why this occurs.**
>
> Thanks for the comment. DyGFormer is the top performer in all the baselines. It achieves state-of-the-art performance by leveraging the attention mechanism across almost all (up to 99%) historical interactions of a node to learn its node embedding. However, this comes at the cost of high GPU memory consumption and slow convergence speed. As demonstrated in Section 4.6, when integrating DyGFormer into our TAMI framework, we are not only able to reduce its GPU memory usage (by 53.22%) and improve its convergence speed (by 40.54%) by leveraging much fewer (16$\times$ fewer) historical interactions of a node, but still improve the performance compared with its vanilla version. This is thanks to LTE, our effective time encoding function to mitigate the skewness of the temporal differences, and LHA for explicit encoding of interaction history of the target node pairs in temporal link prediction.
>
> We also present the ablation study for the historical (hist) and the inductive (ind) negative sampling strategies in the following table. The results confirm the effectiveness of both modules.
>
> **Table 1: Test AP for transductive link prediction.**
> | NSS  | Methods    | CP    | EN     | LA     | UC     |
> |------|------------|-------|--------|--------|--------|
> | hist | DyGFormer  | 97.00 | 75.63  | 81.57  | 82.17  |
> |      | + LTE      | 98.94 | 76.67  | 81.76  | 83.46  |
> |      | + LHA      | 97.63 | 80.99  | 81.96  | 84.35  |
> |      | **+ TAMI** | **98.96** | **81.02** | **83.40** | **85.89** |
> | ind  | DyGFormer  | 95.44 | 77.41  | 73.97  | 72.25  |
> |      | + LTE      | 96.46 | 77.54  | 73.89  | 77.40  |
> |      | + LHA      | 97.16 | 86.12  | 75.02  | 76.86  |
> |      | **+ TAMI** | **97.25** | **86.23** | **74.03** | **80.13** |
>
> - **Q1. [1] has found simple linear time encoding functions to be more effective than the sinusoidal function used in your work. Can the linear time encoding function benefit from your module?**
>
> Thanks for the comment. Yes, the linear time encoding function proposed in [1] can also benefit from our LTE module. Although the linear time encoding function differs from the sinusoidal one, it still takes the raw temporal difference $\Delta t$ as input. We observe that these temporal differences are often right skewed, and TGNNs trained on the skewed input struggle with the link prediction for target node pairs that interact intermittently. By applying the logarithmic transformation to $\Delta t$ using LTE, the skewness of $\Delta t$ can be reduced, resulting in a more balanced input distribution. We have also conducted extra experiments using the linear time encoding function from [1] with LTE. Results in Table 2 below show that with LTE, the performance consistently improves.
>
> **Table 2: Test AP of TGNNs using the linear time encoding function from [1] and the enhanced version with our LTE module. Negative links are generated using the random negative sampling strategy.**
> |Method \ Datasets|uci (UC) |enron (EN)|
> |-|-|-|
> |TGAT (Linear)| 95.41 | 82.31 |
> |**TGAT (Linear + LTE)**| **95.90** | **84.07** |
> |DyGFormer (Linear)| 96.00 | 93.29 |
> |**DyGFormer (Linear + LTE)**| **97.14** | **93.59** |
>
> [1] Chung et al. Between Linear and Sinusoidal: Rethinking the Time Encoder in Dynamic Graph Learning. TMLR
>
> - **Q2. LHA module maintains historical interactions for a target node pair that may be missed when performing temporal node message passing. Why does this approach beat RNN-like methods (e.g., TGN) that can also track very long distance interactions?**
>
> Thanks for the comment. In real-world temporal graphs, each node often interacts with a large number of other nodes, resulting in a long interaction history, or a long sequence, per node. For instance, in the UV dataset, each node interacts with over 5,000 nodes on average, forming a per-node sequence of average length longer than 5,000. However, RNNs are usually not able to track such a long sequence. Furthermore, RNN-based TGNNs still learn each node's temporal embedding based on its recent interactions. Hence, infrequent interactions are still likely to be forgotten. This forgetting problem is also empirically observed in RNN-based TGNNs, as detailed in Appendix C.5. In contrast, our LHA module explicitly encodes the historical interactions between each target node pair using dedicated historical edge embeddings and leverages these edge embeddings together with the node embeddings for the link prediction of each target node pair, thereby mitigating the forgetting problem and improving the link prediction performance.
>
> - **Q2 (cont'd). How does LHA compare to DyG2Vec [2] which also uses that past interactions for a target node pair to build temporal embeddings?**
>
> Thanks for the comment. We respectfully disagree that DyG2Vec utilizes the past interactions between a target node pair to build temporal embeddings. Instead, DyG2Vec, similar to other TGNNs such as GraphMixer and DyGFormer, learns temporal node embeddings based on each node's individual interaction history. Suppose we are to predict the link between node $u$ and node $v$. In DyG2Vec, it considers the $m$-hop neighbors (i.e., past interactions in $m$ time steps) of $u$ to learn its node embedding, together with the edge embeddings from these $m$-hop neighbors. Similarly for node $v$. However, the resulting node embeddings may not capture the interaction between $u$ and $v$, as $v$ may not be in the $m$-hop neighborhood of $u$ (and vice versa). In contrast, our LHA directly encodes the past interactions between $u$ and $v$ using dedicated historical edge embeddings, and leverages this edge embedding explicitly for the link prediction between them.
>
> [2] Alomrani et al. DyG2Vec: Efficient Representation Learning for Dynamic Graphs. TMLR
>
> - **Q3. Does the choice of aggregation function for LHA affect performance?**
>
> Thanks for the comment. We have conducted extra experiments using different aggregation functions with varying values of $k$, where $k$ is the number of historical edge embeddings used in Equation (5) of our manuscript. The results presented in Table 3 below show that LHA is robust to the choice of aggregation functions.
>
> **Table 3: Test AP of DyGFormer on the EN and CP datasets with different $k$ values and aggregation strategies.**
> | Method | EN | CP |
> |-|-|-|
> | DyGFormer (Vanilla) | 92.46 | 97.91 |
> | w/ TAMI             |       |       |
> | Most-recent         | 92.66 | 98.67 |
> | Mean ($k=1$)        | 92.66 | 98.67 |
> | Mean ($k=2$)        | 92.60 | 98.76 |
> | **Mean ($k=3$)**        | 92.56 | **98.77** |
> | Max ($k=1$)         | 92.66 | 98.67 |
> | **Max ($k=2$)**         | **92.84** | 98.74 |
> | Max ($k=3$)         | 92.60 | 98.74 |
>
> - **Possible typo in figure 4: According to your notation $t5$ should be the oldest timestep while $t1$ is the latest timestep but the figure suggests otherwise**
>
> Thanks for pointing this out. We have updated accordingly.

---

> > ### Author Response · Authors · 2025-08-04
> >
> > Dear Reviewer,
> >
> > Thank you again for recognizing the significance of our proposed framework and the comprehensiveness of our experiments.
> >
> > We have provided detailed explanations and additional experiment results in response to your questions.
> >
> > Based on your suggestions, we will correct the typo in Figure 4 and add the corresponding experiments and discussions to our revised manuscript.
> >
> > As the discussion period is halfway through, we would greatly appreciate your feedback on these responses to confirm that they fully meet your expectations. Please kindly let us know if you have any further questions. We will be very happy to answer.
> >
> > Sincerely,
> >
> > Authors

---

> > ### Author Response · Authors · 2025-08-08
> >
> > Dear Reviewer,
> >
> > Thank you very much for reviewing our paper. We hope our responses have addressed your concerns, and we sincerely appreciate that you can recommend accepting our paper. Please also let us know if you have any further questions, we will be very happy to answer.
> >
> > Best Regards,
> >
> > Authors

---

### Decision · Program_Chairs · 2025-09-17

**Decision:**

Accept (poster)

**Comment:**

This paper identifies that skewed interaction patterns in temporal graphs hurt link prediction and proposes TAMI, a lightweight framework that uses logarithmic time encoding (LTE) to balance temporal data and link history aggregation (LHA) to prevent models from forgetting past, infrequent interactions.

The paper's primary strength is its novel identification of the interaction heterogeneity problem, supported by an exceptionally rigorous experimental evaluation that validates its simple, effective, and broadly applicable solution. Its main weaknesses are the limited technical novelty of its components and its unproven applicability beyond the link prediction task, with performance gains being modest on some top-performing models.

In the discussion, the authors addressed the main weaknesses to some extent. Initial concerns about limited novelty and an insufficient set of modern baselines were partially resolved when the authors provided new experiments integrating TAMI and conducted a more comprehensive efficiency analysis, convincing the reviewers to raise their score. Similarly, questions about inconsistent performance gains on certain datasets were clarified with deeper data analysis that explained the observed phenomena. The only remaining point is the framework's applicability beyond link prediction, which the authors acknowledged as a valid limitation and a direction for future work.

I also noticed the limited broad usage of the proposed techniques. But the extensive experiments and analysis outweigh the weaknesses. Therefore, I suggest accepting the paper as a poster,